# Starvation-induced regulation of carbohydrate transport at the blood–brain barrier is TGF-β-signaling dependent

Helen Hertenstein[1], Ellen McMullen[2], Astrid Weiler[1], Anne Volkenhoff[1], Holger M Becker[1,3], Stefanie Schirmeier[1]*

[1]Department of Biology, Institute of Zoology, Technische Universität Dresden, Dresden, Germany; [2]Institut für Neuro- und Verhaltensbiologie, WWU Münster, Münster, Germany; [3]Division of General Zoology, Department of Biology, University of Kaiserslautern, Kaiserslautern, Germany

**Abstract** During hunger or malnutrition, animals prioritize alimentation of the brain over other organs to ensure its function and, thus, their survival. This protection, also-called brain sparing, is described from *Drosophila* to humans. However, little is known about the molecular mechanisms adapting carbohydrate transport. Here, we used *Drosophila* genetics to unravel the mechanisms operating at the blood–brain barrier (BBB) under nutrient restriction. During starvation, expression of the carbohydrate transporter Tret1-1 is increased to provide more efficient carbohydrate uptake. Two mechanisms are responsible for this increase. Similar to the regulation of mammalian GLUT4, Rab-dependent intracellular shuttling is needed for Tret1-1 integration into the plasma membrane; even though Tret1-1 regulation is independent of insulin signaling. In addition, starvation induces transcriptional upregulation that is controlled by TGF-β signaling. Considering TGF-β-dependent regulation of the glucose transporter GLUT1 in murine chondrocytes, our study reveals an evolutionarily conserved regulatory paradigm adapting the expression of sugar transporters at the BBB.

**\*For correspondence:**
stefanie.schirmeier@tu-dresden.de

**Competing interests:** The authors declare that no competing interests exist.

## Introduction

A functional nervous system is essential for an animal's survival. To properly function, the nervous system needs a disproportionately large amount of energy relative to its size. The human brain for example accounts for only about 2% of the body's weight but uses around 20% of the resting oxygen consumption (*Laughlin et al., 1998*). Likewise, the insect retina consumes approximately 10% of the total ATP generated (*Harris et al., 2012*; *Laughlin et al., 1998*; *Mink et al., 1981*).

The nervous system is very susceptible to changing extracellular solute concentrations and thus needs to be separated from circulation. This task is performed by the blood–brain barrier (BBB), which prevents paracellular diffusion, and thereby uncontrolled influx of ions, metabolites, xenobiotics, pathogens, and other blood-derived potentially harmful substances. Protein, ion, and metabolite concentrations fluctuate to a much greater extent in circulation than in the cerebrospinal fluid, the brains extracellular milieu (*Begley, 2006*). Thus, fluxes over the BBB must be tightly regulated and only small lipid-soluble molecules and gases like $O_2$ and $CO_2$ can diffuse freely (*van de Waterbeemd et al., 1998*).

The enormous energy demand of the nervous system is mainly met by carbohydrate metabolism. The human brain takes up approximately 90 g glucose per day during adulthood and up to 150 g per day during development (*Kuzawa et al., 2014*). Since glucose and other carbohydrates are hydrophilic molecules, free diffusion over the BBB is impossible. Therefore, carbohydrates need to be transported into the nervous system via specialized transport proteins. In mammals, glucose

transporter 1 (GLUT1, encoded by the *Slc2a1 [solute carrier family 2 member 1]* gene) is considered to be the main carbohydrate transporter in the BBB-forming endothelial cells. Aberrations in carbohydrate availability or transport are thought to be a major factor in the development of diverse neurological diseases such as GLUT1 deficiency syndrome, Alzheimer's disease, and epilepsy (*Arsov et al., 2012*; *Hoffmann et al., 2013*; *Kapogiannis and Mattson, 2011*; *Koepsell, 2020*). Therefore, understanding the regulatory mechanisms that govern carbohydrate transport into the nervous system is of major interest. Interestingly, it has been reported that endothelial GLUT1 expression is increased upon hypoglycemia (*Boado and Pardridge, 1993*; *Kumagai et al., 1995*; *Simpson et al., 1999*, reviewed in *Patching, 2017*; *Rehni and Dave, 2018*). However, the molecular mechanisms that control this upregulation are not yet understood. In addition, upon oxygen or glucose deprivation, that are a consequence of ischemia, expression of the sodium glucose cotransporters SGLT1 (*Slc5a1*) and SGLT2 (*Slc5a2*) is induced in brain endothelial cells (*Elfeber et al., 2004*; *Enerson and Drewes, 2006*; *Nishizaki et al., 1995*; *Nishizaki and Matsuoka, 1998*; *Vemula et al., 2009*; *Yu et al., 2013*). Overall, this indicates that carbohydrate transport at the BBB can adapt to changes in carbohydrate availability in various ways. However, the molecular underpinnings of the different regulatory processes remain elusive.

As it is the case in vertebrates, the insect nervous system must be protected by a BBB. Since insects have an open circulatory system, the brain is not vascularized but is surrounded by the blood-like hemolymph. In *Drosophila*, the BBB surrounds the entire nervous system to prevent uncontrolled entry of hemolymph-derived substances. It is formed by two glial cell layers, the outer perineurial and inner subperineurial glial cells (reviewed in *Limmer et al., 2014*; *Yildirim et al., 2019*). The *Drosophila* BBB shares fundamental functional aspects with the vertebrate BBB. The subperineurial glial cells build a diffusion barrier by forming intercellular pleated septate junctions that prevent paracellular diffusion (*Stork et al., 2008*). In addition, efflux transporters export xenobiotics and many solute carrier family transporters supply the brain with essential ions and nutrients (*DeSalvo et al., 2014*; *Hindle and Bainton, 2014*; *Lane and Treherne, 1972*; *Mayer and Belsham, 2009*; *Stork et al., 2008*; reviewed in *Weiler et al., 2017*). In the *Drosophila* hemolymph, in addition to glucose, trehalose, a non-reducing disaccharide consisting of two glucose subunits linked by an $\alpha,\alpha-1,1$-glycosidic bond, is found in high quantities. Fructose is also present, albeit in low and highly fluctuating concentrations, making its nutritional role questionable (*Blatt and Roces, 2001*; *Broughton et al., 2008*; *Lee and Park, 2004*; *Pasco and Léopold, 2012*; *Wyatt and Kalf, 1957*). Transcriptome data of the BBB-forming glial cells suggests expression of several putative carbohydrate transporters (*DeSalvo et al., 2014*; *Ho et al., 2019*). The closest homologs of mammalian GLUT1–4 are dmGlut1, dmSut1, dmSut2, dmSut3, and CG7882. dmGlut1 has been shown to be expressed exclusively in neurons (*Volkenhoff et al., 2018*). In situ, microarray and single-cell sequencing data indicate very low or no expression for dmSut1-3 and CG7882 in the nervous system (*Croset et al., 2018*; *Davie et al., 2018*; *Weiszmann et al., 2009*). The carbohydrate transporter Tret1-1 (Trehalose transporter 1–1) is specifically expressed in perineurial glia (*Volkenhoff et al., 2015*). Tret1-1 is most homologous to mammalian GLUT6 and GLUT8 and has been shown to transport trehalose when heterologously expressed in *Xenopus laevis* oocytes (*Kanamori et al., 2010*).

The *Drosophila* nervous system, as the mammalian nervous system, is protected from growth defects caused by malnutrition through a process called brain sparing. It has been shown that Jelly belly (Jeb)/anaplastic lymphoma kinase (ALK) signaling constitutes an alternative growth-promoting pathway active in neuroblasts (neuronal stem cells) allowing their continuous division (reviewed in *Lanet and Maurange, 2014*; *Cheng et al., 2011*). However, if the brain continues developing and keeps its normal function, nutrient provision needs to be adapted to ensure sufficient uptake, even under challenging circumstances, like low circulating carbohydrate levels. How nutrient transport at the BBB is adapted to meet the needs of the nervous system even under nutrient restriction has not yet been studied.

Here, we show that carbohydrate transporter expression in *Drosophila*, as in mammals, adapts to changes in carbohydrate availability in circulation. Tret1-1 expression in perineurial glia of *Drosophila* larvae is strongly upregulated upon starvation. This upregulation is triggered by starvation-induced hypoglycemia as a mechanism protecting the nervous system from the effects of nutrient restriction. Ex vivo glucose uptake measurements using a genetically encoded Förster resonance energy transfer (FRET)-based glucose sensor show that the upregulation of carbohydrate transporter expression leads to an increase in carbohydrate uptake efficiency. The compensatory upregulation

of Tret1-1 transcription is independent of insulin/adipokinetic hormone signaling, but instead depends on TGF-β signaling. This regulatory mechanism that protects the brain from the effects of malnutrition is likely conserved in mammals, since mammalian Glut1 is also upregulated in the BBB upon hypoglycemia and has been shown to be induced by TGF-β signaling in other tissues (*Boado and Pardridge, 1993*; *Kumagai et al., 1995*; *Simpson et al., 1999*; *Lee et al., 2018*).

## Results

### Tret1-1 is upregulated in perineurial glial cells upon starvation

The *Drosophila* larval brain is separated from circulation by the BBB to avoid uncontrolled leakage of hemolymph-derived potentially harmful substances. At the same time, the blood-brain barrier also separates the brain from nutrients available in the hemolymph. Thus, transport systems are necessary to ensure a constant supply of nutrients, including carbohydrates. The trehalose transporter Tret1-1 is expressed in the perineurial glial cells of the larval and adult nervous system (*Volkenhoff et al., 2015*). In order to better understand whether carbohydrate transport at the BBB is adapted to the metabolic state of the animal, we analyzed Tret1-1 dynamics under different physiological conditions. To do so we subjected larvae to chronic starvation applying a well-established paradigm that allows 40 hr of starvation without disturbing development (*Figure 1A*, *Zinke et al., 2002*). Seventy hour AEL larvae undergo an organismal change that allows their survival even under complete nutritional restriction (*Beadle et al., 1938*). Therefore, we starved animals before this time-point to study the importance of Tret1-1 in a nutrient-dependent manner. In fed animals, Tret1-1 can be found at the plasma membrane of the perineurial glial cells (*Figure 1B–E*, *Volkenhoff et al., 2015*). However, a large portion of the protein localizes to intracellular vesicles (dotted structure in *Figure 1E*, *Figure 2—figure supplement 1*, *Volkenhoff et al., 2015*). Starvation increases Tret1-1 protein levels in perineurial glial cells (compare *Figure 1D,E–D',E'*). Furthermore, more Tret1-1 protein can be found at the plasma membrane (*Figure 1D',E'*, arrows, *Figure 2—figure supplement 1*). Whether the proportion of Tret1-1 at the plasma membrane is increased or if the increase in Tret1-1 at the plasma membrane is due to the general increase in Tret1-1 protein remains unclear.

### Intracellular trafficking of Tret1-1 is Rab7 and Rab10 dependent

Three mammalian glucose transporters, GLUT4, GLUT6, and GLUT8, are regulated via trafficking between storage vesicles and the plasma membrane (*Corvera et al., 1994*; *Cushman and Wardzala, 1980*; *Lisinski et al., 2001*; *Suzuki and Kono, 1980*). Similarly, a large amount of Tret1-1 localizes to intracellular vesicles (*Figure 1E*, *Figure 2—figure supplement 1*). Thus, intracellular trafficking of Tret1-1 may partially regulate carbohydrate uptake into the perineurial glial cells.

To analyze whether regulation of Tret1-1 expression requires intracellular trafficking, we studied the involvement of different Rab-GTPases. Utilizing an EYFP-Rab library available for *Drosophila* (*Dunst et al., 2015*), we found that subsets of Tret1-1-positive vesicles are also positive for Rab7, Rab10, Rab19, and Rab23 (*Figure 2—figure supplement 1*). Rab7 is needed for the formation of late endosomes and their fusion with lysosomes, while Rab10 has been implicated in GLUT4 storage vesicle trafficking in mammals (reviewed in *Guerra and Bucci, 2016*; *Huotari and Helenius, 2011*; *Klip et al., 2019*). The roles of Rab19 and Rab23 are less well understood. Rab23 has been implicated in planar cell polarity and in Hedgehog regulation in response to dietary changes, but its exact functions are unclear (*Çiçek et al., 2016*; *Pataki et al., 2010*). Rab19 has been described to act in enteroendocrine cell differentiation, but its role in this process is unknown (*Nagy et al., 2017*).

To determine a possible functional role of these Rab-GTPases in regulating Tret1-1 trafficking, we analyzed Tret1-1 localization in the background of a glia-specific knockdown (or expression of dominant-negative forms) of the respective Rab proteins (*Figure 2A–G*). Silencing of Rab19 or Rab23 did not induce any misregulation or mislocalization of Tret1-1 in perineurial glial cells (data not shown). In contrast, interfering with Rab7 or Rab10 function induced distinct abnormal phenotypes (*Figure 2*). Panglial knockdown of Rab7 using RNA interference reduced the levels of Tret1-1 (*Figure 2B,B'*). This phenotype was also observed when a dominant-negative form of Rab7, Rab7$^{T22N}$ was expressed in all glia (*Figure 2F,F'*). The dominant-negative Rab-constructs used here are tagged with an N-terminal YFP and thus induce a weak background staining in all glial cells (*Figure 2F,G*, asterisks). The effect of Rab seven knockdown on Tret1-1 was further verified by BBB-specific

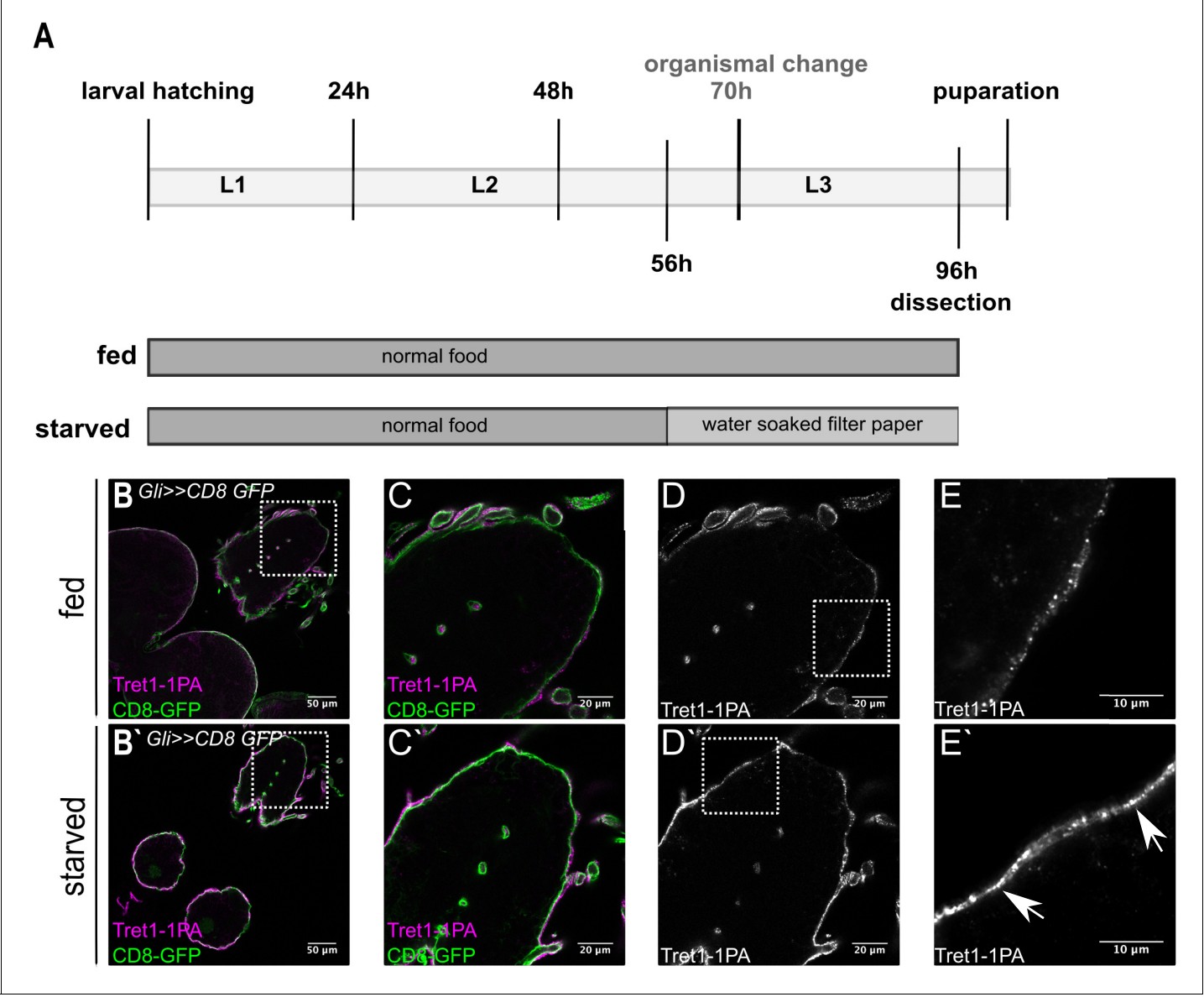

**Figure 1.** Tret1-1 expression is upregulated upon starvation. (**A**) Scheme of the starvation paradigm. Fed animals were kept for 96 hr on normal food before dissecting. Starved animals were transferred onto a water-soaked filter paper 56 hr after larval hatching. Forty hour later these animals were also dissected and immunohistochemistry was performed. (**B–E**) Brains of fed larvae expressing CD8-GFP in the subperineurial glial cells (Gli>>CD8-GFP) stained for GFP (green) and Tret1-1 (magenta/gray). (**B'–E'**) Brains of starved larvae with the same genotype. (**D, D'**) Tret1-1 expression in the perineurial glial cells is induced upon starvation. (**E, E'**) Close up of the BBB. Tret1-1 is localized in vesicles and its expression is elevated upon starvation. Tret1-1 is localized to the plasma membrane (arrows).

knockdown of Rab7 (*Figure 2I*). The reduced Tret1-1 level in Rab7 loss of function indicates that blocking late endosome to lysosome maturation and thus possibly blocking Tret1-1 degradation, induces a negative feedback that reduces Tret1-1 expression.

In contrast to Rab7, knockdown of Rab10 in all glia, or in the BBB-glial cells specifically, leads to a prominent accumulation of Tret1-1 in the perineurial cytosol (*Figure 2D,D', J*). This phenotype was reproduced when a dominant-negative form of Rab10, Rab10$^{T23N}$, was expressed in glial cells (*Figure 2G,G'*). This suggests a major role of Rab10 in delivering Tret1-1 to the plasma membrane of perineurial glial cells. In summary, Tret1-1 homeostasis is dependent on Rab-GTPase-mediated intracellular trafficking.

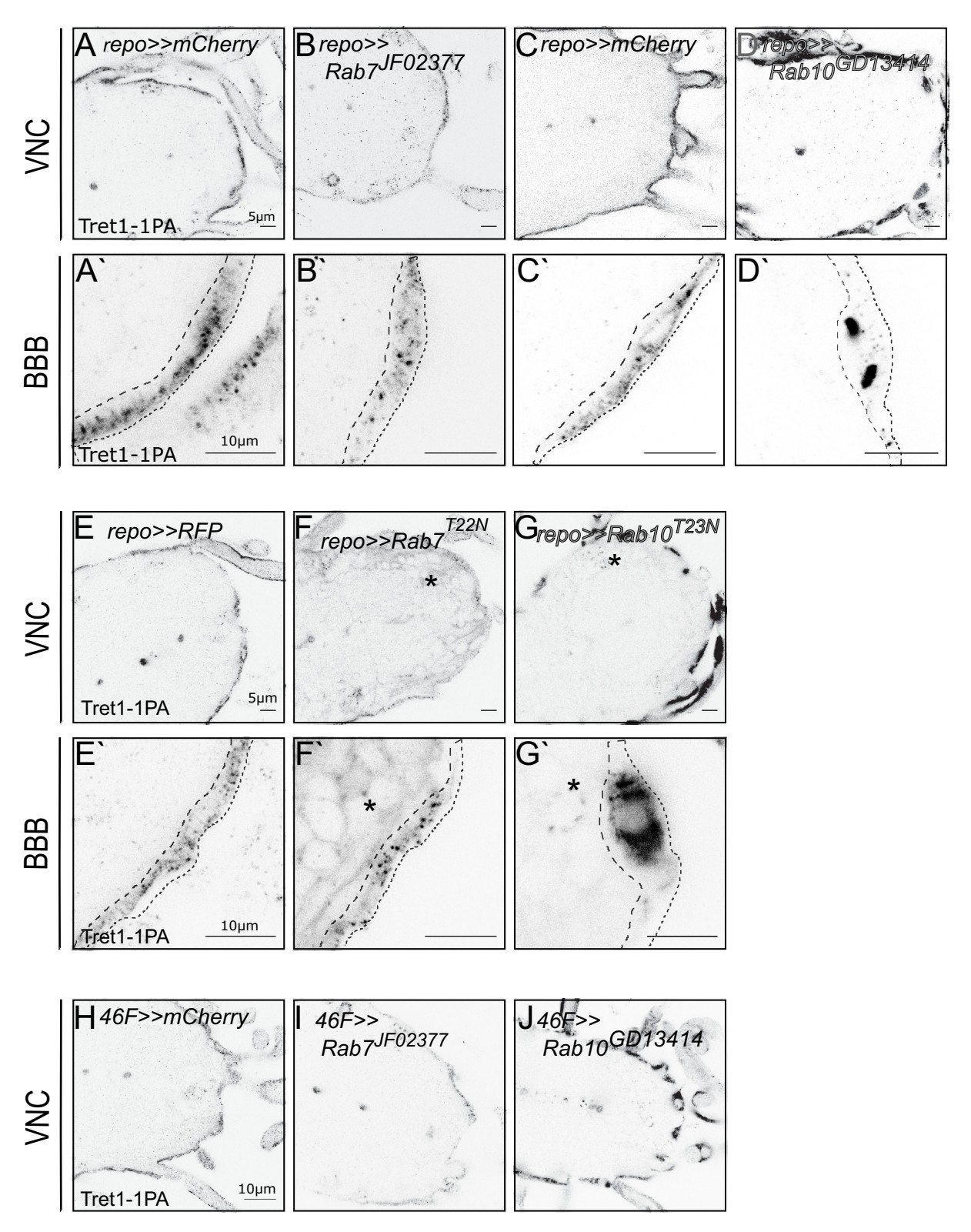

**Figure 2.** Tret1-1 intracellular trafficking depends on Rab7 and Rab10. (A, C) Tret1-1 staining of the ventral nerve cord of glia-specific (repo-Gal4) knockdowns (A', B', C', D') shows a close up of the BBB. Dotted lines show the outline of the perineurial glia. (B, B') Tret1-1 expression is strongly reduced by a glial Rab7 (*Rab7*$^{JF02377}$) knockdown. (D, D') Disrupting Rab10 expression in glia (*Rab10*$^{GD13414}$) induces accumulation of Tret1-1 in the perineurial glia cytosol. (E–G') Glial expression of the dominant-negative constructs *Rab7*$^{T22N}$ and *Rab10*$^{T23N}$ induce similar phenotypes as the RNA

*Figure 2 continued on next page*

Figure 2 continued

interference mediated knockdowns. (F, F') Expressing $Rab7^{T22N}$ reduces Tret1-1 staining. (G, G') Glia expression of $Rab10^{T23N}$ induces transporter mislocalization and a strong accumulation in the perineurial cytosol. (F–G') The dominant-negative Rab-constructs are Rab-YFP fusions. Panglial overexpression thus leads to a weak background staining in the green channel (asterisks). (C) Tret1-1 staining of surface and cortex glia-specific knockdown using *46 F-Gal4* and $Rab7^{JF02377}$ and $Rab10^{GD13414}$. Loss of Rab7 reduces Tret1-1 staining, while Rab10 disruption induces transporter mislocalization.

The online version of this article includes the following figure supplement(s) for figure 2:

**Figure supplement 1.** Rab7, Rab10, Rab19, and Rab23 colocalize with Tret1-1 vesicles.

## Tret1-1 is transcriptionally regulated upon starvation

To test whether transcriptional regulation accounts for the strong increase in Tret1-1 protein upon starvation, we cloned the *Tret1-1* promotor and established transgenic animals expressing either Gal4 or a nuclear GFP (stinger-GFP, stgGFP) under its control (*Figure 3—figure supplement 1*). We validated the expression induced by the promotor fragment by co-staining RFP expressed under *Tret1-1-Gal4* control with the Tret1-1 antibody we generated previously (*Volkenhoff et al., 2015*). *Tret1-1* promotor expression and Tret1-1 protein colocalize well in the nervous system (*Figure 3—figure supplement 1B*, arrows). We previously showed that Tret1-1 localizes to perineurial glial cells and some unidentified neurons (*Figure 3—figure supplement 1B* stars; *Volkenhoff et al., 2015*). To verify the perineurial glial expression of the *Tret1-1* promotor, we stained *Tret1-1-stgGFP* animals for a nuclear perineurial glial marker, Apontic (*Figure 3—figure supplement 1C*, *Zülbahar et al., 2018*). Apontic and stgGFP colocalize in perineurial nuclei (*Figure 3—figure supplement 1C*, stars).

To analyze changes in *Tret1-1* transcription levels, we subjected animals expressing stgGFP under the control of the *Tret1-1* promotor to our starvation paradigm. Starvation induces a robust increase of stgGFP in the brains of starved larvae as quantified by western blot (*Figure 3A,A'*). To verify the upregulation of the *Tret1-1* promotor, we additionally quantified stgGFP fluorescence in brain stainings. We normalized the GFP signal of individual nuclei to 4',6-diamidino-2-phenylindole (DAPI), since it is highly unlikely that the DNA content of the nuclei would change upon starvation. The level of stgGFP expressed under *Tret1-1* control is significantly higher in brains of starved animals compared to control larvae (*Figure 3B*). These experiments show that the *Tret1-1* promotor is induced upon starvation and thus Tret1-1 levels are transcriptionally adapted to the animal's metabolic state.

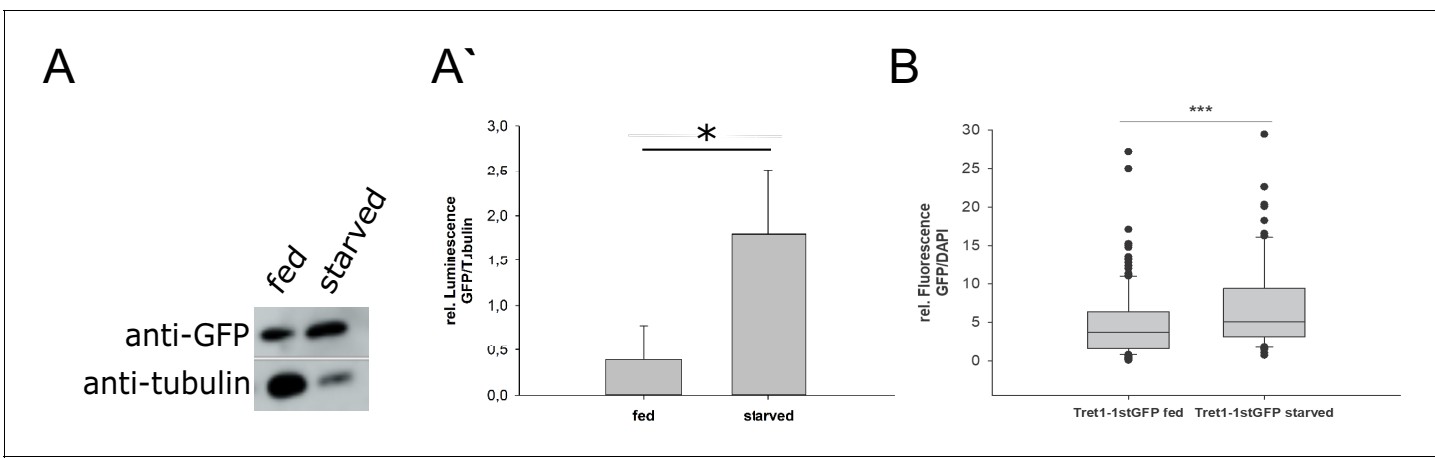

**Figure 3.** Tret1-1 is transcriptionally upregulated upon starvation. (A) Tret1-1 is transcriptionally upregulated upon starvation since the Tret1-1 reporter *Tret1-1>stgGFP* is significantly upregulated in brains of starved compared to fed larvae as seen in western blots. Shown are images of representative western blots for anti-GFP and anti-tubulin (loading control). n=3 (A') Quantification of Western blots. N=3. (B) Tret1-1 protein is significantly upregulated upon starvation in *Tret1-1>stgGFP* animals. Quantified is the GFP fluorescence normalized to DAPI in individual nuclei. N=5 , n > 34.

The online version of this article includes the following figure supplement(s) for figure 3:

**Figure supplement 1.** Tret1-1 promoter drives specific expression.

## Increase in Tret1-1 expression upon starvation is sugar dependent

The expression of mammalian GLUT1 in brain endothelial cells increases upon chronic hypoglycemia (*Boado and Pardridge, 1993*; *Kumagai et al., 1995*; *Rehni and Dave, 2018*; *Simpson et al., 1999*). In *Drosophila*, starvation results in hypoglycemia (*Dus et al., 2011*; *Matsuda et al., 2015*). Thus, we wondered if the increase in Tret1-1 protein levels described here might be induced by a reduction in circulating carbohydrate levels. To understand whether dietary carbohydrates are sufficient to circumvent Tret1-1 induction, we compared animals fed on standard food (*Figure 4A*), starved animals (*Figure 4A'*), and animals fed on 10% sucrose in phosphate-buffered saline (*Figure 4A''*). Tret1-1 fluorescent intensity increases two to three times upon starvation compared to fed animals (*Figure 4B*). However, larvae kept on sugar-only food do not display increased Tret1-1 levels, but levels comparable to larvae kept on standard food (*Figure 4A,A', A''*). The ratio of Tret1-1 intensity of animals kept on sugar food versus fed animals is around 1, indicating no increase in Tret1-1 signal (*Figure 4B*). Hence, dietary sugar abolishes Tret1-1 induction, indicating that other nutrients, like amino acids, are not important for Tret1-1 upregulation. Attempts to analyze Tret1-1 levels in larvae fed on a protein-only diet to study the influence of dietary amino acids were unsuccessful as larvae do not eat protein-only diet (no uptake of colored protein-only food into the intestine over 48 hr, data not shown). This data suggests that Tret1-1 is upregulated in the perineurial glial cells upon starvation-induced hypoglycemia. Hence, the Tret1-1 promoter fragment does include a starvation-sensitive motive.

## Glucose uptake rate increases upon starvation

Tret1-1 upregulation in perineurial glial cells is most likely a mechanism that ensures efficient carbohydrate uptake into the nervous system even under conditions of low circulating carbohydrate levels. Therefore, we aimed to study the impact of Tret1-1 upregulation on carbohydrate uptake. *Kanamori et al., 2010* showed that Tret1-1 transports trehalose when heterologously expressed in *X. laevis* oocytes. Since not only trehalose, but also glucose and fructose, are found in the *Drosophila* hemolymph, we analyzed whether Tret1-1 also transports other carbohydrates. We expressed Tret1-1 in *X. laevis* oocytes to study its substrate specificity. The Tret1-1 antibody is specific to the Tret1-1 PA isoform, and thus at least this isoform is upregulated in the perineurial glial cells upon starvation. Therefore, we expressed a 3xHA-tagged version of Tret1-1PA in *X. laevis* oocytes. The functionality of this construct was verified by its ability to rescue the lethality associated with *Tret1-1$^{-/-}$* mutants when ubiquitously expressed (using *da-Gal4*, *Volkenhoff et al., 2015*). Incubating *X. laevis* oocytes expressing Tret1-1PA-3xHA with different concentrations of $^{14}C_6$-fructose, $^{14}C_6$-glucose, or $^{14}C_{12}$-trehalose for 60 min, we were able to verify the trehalose transport capacity reported previously

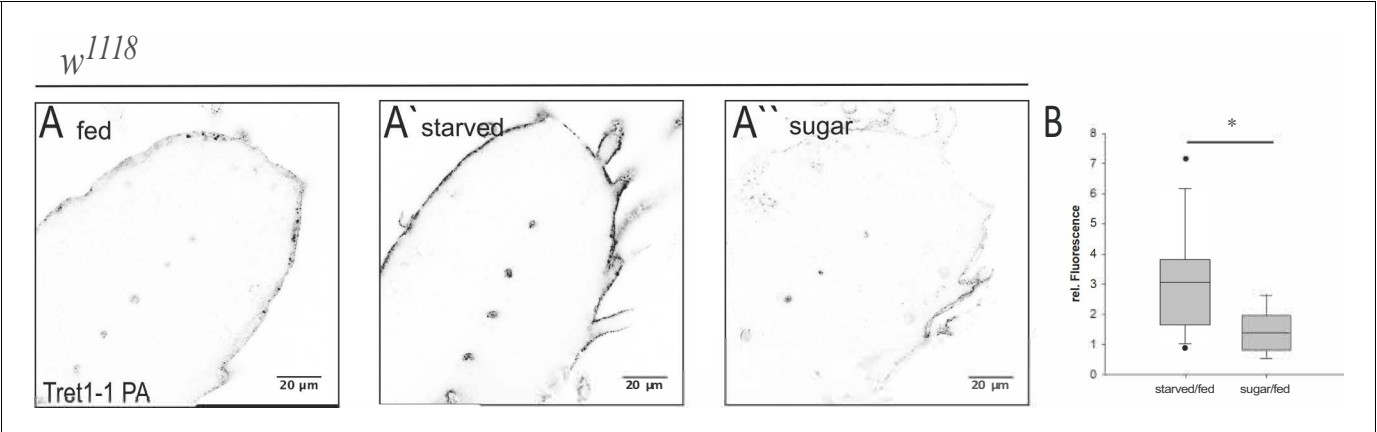

**Figure 4.** Tret1-1 upregulation upon starvation is sugar-dependent. (A- A'') Brains of larvae kept on normal food, under starvation conditions or on sugar food (10% sucrose) were stained for Tret1-1. (A') Tret1-1 expression is elevated upon starvation of the animal. (A'') Dietary sugar reverses Tret1-1 upregulation completely. (B) Quantification of Tret1-1 expression in starved wild type animals and wild type animals on sugar food. The quantification shows the ratio of relative Tret1-1 fluorescence intensity in the perineurial glial cells of starved versus fed and sugar-fed vs. fed animals. N≥4; n=9-15; *p<0,05.

(*Kanamori et al., 2010*, *Figure 5A*). In addition, Tret1-1PA can facilitate uptake of glucose, while fructose is not taken up efficiently (*Figure 5A*).

Taking advantage of the glucose transport capacity of Tret1-1, we employed the Förster resonance energy transfer (FRET)-based glucose sensor FLII[12]Pglu-700μδ6 (*Takanaga et al., 2008*; *Volkenhoff et al., 2018*) to determine the effect of Tret1-1 upregulation on carbohydrate import into the living brain. A trehalose sensor to measure trehalose uptake is unfortunately not available. However, the glucose sensor allows live imaging of glucose uptake in a cell type of choice in ex vivo brain preparations (*Volkenhoff et al., 2018*). We expressed FLII[12]Pglu-700μδ6 specifically in the BBB-glial cells (*9137*-Gal4, *DeSalvo et al., 2014*). The respective larvae were subjected to the starvation protocol, and, subsequently, glucose uptake was measured (*Figure 5D–F*). The rate of glucose uptake was significantly increased in brains of starved animals compared to the brains of age-matched animals kept on standard food (*Figure 5D,E*). We now asked whether this elevated glucose uptake upon starvation is specifically caused by Tret1-1 upregulation. Therefore, we knocked down

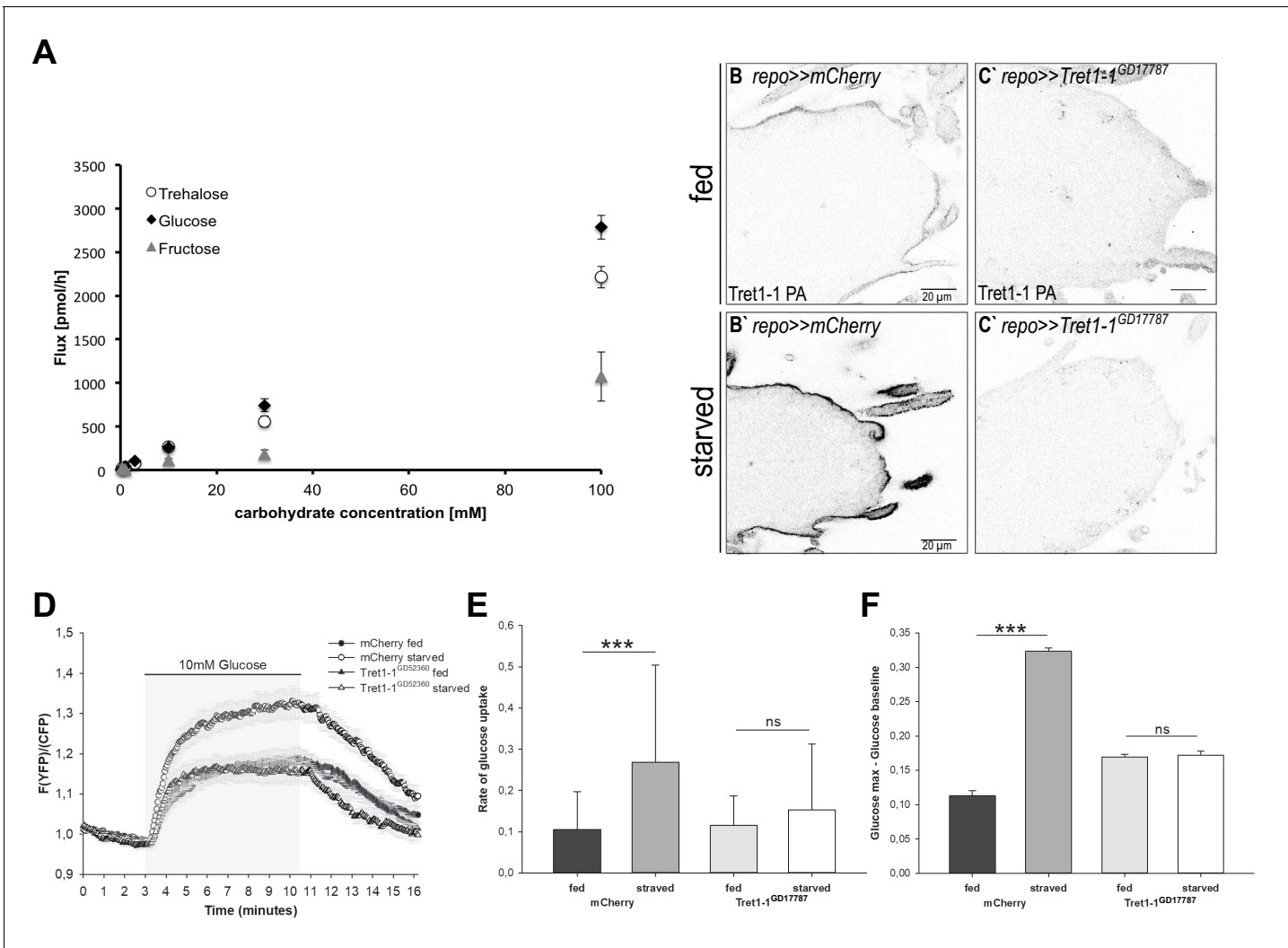

**Figure 5.** Carbohydrate uptake rate into the surface glia is elevated upon starvation. (A) Carbohydrate uptake capacity of *Xenopus laevis* oocytes heterologously expressing Tret1-1PA-3xHA. Tret1-1PA facilitates uptake of glucose and trehalose. In contrast, fructose uptake rate is minor. (B–C') Expressing *Tret1-1*[GD17787] in glial cells induces a loss of the specific Tret1-1 staining in perineurial glia. (C') No increase upon starvation can be detected. (D) Glucose uptake was measured in ex vivo brains of fed or starved larvae using the genetically encoded glucose sensor FLII[12]Pglu-700μδ6. Shown are mean traces (n = 10). Error bars are standard error. (E) The glucose uptake rate is significantly higher in brains of starved control larvae compared to fed control larvae. Knocking down Tret1-1 prohibits the increased glucose uptake upon starvation. (F) In addition, the maximum intracellular glucose concentration is significantly higher in starved control larvae than in fed control larvae, suggesting that the uptake rate exceeds the rate of metabolism. This effect is also abolished when Tret1-1 is impaired in BBB-glia. N = 3, n ≥ 10.

Tret1-1 expression in the BBB by expressing *Tret1-1*-dsRNA. Knockdown of *Tret1-1* in glial cells leads to a loss of Tret1-1 staining, verifying the functionality of the construct (*Figure 5C,C'*). We measured glucose uptake into *Tret1-1* knockdown brains. Indeed, the increase in glucose uptake is abolished when Tret1-1 is impaired (*Figure 5E,F*). Interestingly, fed *Tret1-1* knockdown animals show somewhat elevated glucose levels compared to control animals (*Figure 5E*). Two additional sugar transporters, MFS3 and Pippin, have been shown to be expressed in the surface glia (*McMullen et al., 2021*). These transporters might be upregulated upon loss of Tret1-1 and over-compensate glucose transport in fed animals. In summary, these findings show that, indeed, carbo-hydrate uptake into the brain is more efficient in starved animals and that the elevated glucose uptake efficiency is dependent on Tret1-1 expression. Such improved carbohydrate uptake most likely protects the brain from the effects of low circulating carbohydrate levels.

## Starvation-induced upregulation of Tret1-1 is insulin and adipokinetic hormone independent

The plasma membrane localization of mammalian GLUT4 is regulated by insulin (reviewed in *Klip et al., 2019*). Since starvation changes circulating carbohydrate levels, it has strong effects on insulin and adipokinetic hormone (AKH) signaling (reviewed in *Nässel et al., 2015*). Thus, insulin/AKH signaling may control Tret1-1 induction upon starvation. To study the implication of insulin signaling, we expressed dominant-negative forms of the insulin receptor (InR, InR$^{K1409A}$, and InR$^{R418P}$) in the BBB-forming glial cells (*Figure 6A–C*). If Insulin signaling was to directly regulate Tret1-1 tran-scription, one would assume a negative effect, since *Tret1-1* is upregulated upon starvation when insulin levels are low. If insulin signaling indeed has a negative effect on *Tret1-1* expression, higher Tret1-1 levels would be expected under fed conditions upon expression of a dominant-negative InR. Expression of dominant-negative forms of InR did not changed Tret1-1 levels in fed animals in com-parison to the control (*Figure 6A–C,A'-C'*). In addition, Tret1-1 upregulation upon starvation was indistinguishable from that observed in control animals (*Figure 6D*), indicating that Tret1-1 transcrip-tion is independent of insulin signaling.

In *Drosophila*, AKH is thought to play a role equivalent to glucagon/glucocorticoid signaling in mammals (*Gáliková et al., 2015*). AKH signaling induces lipid mobilization and foraging behavior, at least in the adult animal (*Gáliková et al., 2015*). Thus, AKH signaling would be a good candidate to induce *Tret1-1* upregulation upon starvation. We analyzed Tret1-1 levels in *Akh*$^{−/−}$ (*Akh*$^{SAP}$ and *Akh*$^{AP}$) mutant animals under normal conditions and starvation. Tret1-1 levels in the perineurial glial cells in both fed and starved *Akh*$^{−/−}$ mutant larvae are indistinguishable from control levels (*Figure 6E–G,E'–G'*). Interestingly, Tret1-1 is still induced upon starvation in *Akh*$^{−/−}$ mutant animals (*Figure 6H*). This suggests that AKH does not play a role in Tret1-1 regulation upon starvation. In summary, the core signaling pathways regulating organismal nutrient homeostasis, Insulin and AKH signaling, are not involved in Tret1-1 upregulation upon starvation.

## Jeb/ALK signaling does not regulate Tret1-1 expression

Tret1-1 upregulation upon starvation is likely a mechanism to spare the nervous system from the effects of restricted nutrient availability. Jeb/ALK signaling is important to allow continued develop-mental brain growth even upon poor nutrition (*Cheng et al., 2011*). To analyze whether this pathway might also play a role in adapting carbohydrate transport, we knocked down *jeb* and *Alk* in all glial cells and analyzed Tret1-1 expression. *Alk* knockdown in glial cells did not induce changes in Tret1-1 expression compared to control animals (*Figure 6—figure supplement 1*). Tret1-1 is still upregu-lated upon starvation, indicating that ALK signaling in glial cells is not involved in Tret1-1 regulation (*Figure 6—figure supplement 1B,B'*). *jeb* knockdown in all glial cells induced strong starvation sus-ceptibility of the animals in our hands. Most animals died within the 40 hr starvation period, which does not happen to control animals. Analysis of Tret1-1 expression in the perineurial glial cells of escapers did not give coherent results. Nevertheless, since *Alk* knockdown shows wild typic Tret1-1 upregulation, Jeb/ALK signaling is most likely not implicated in the regulation of carbohydrate trans-port upon starvation.

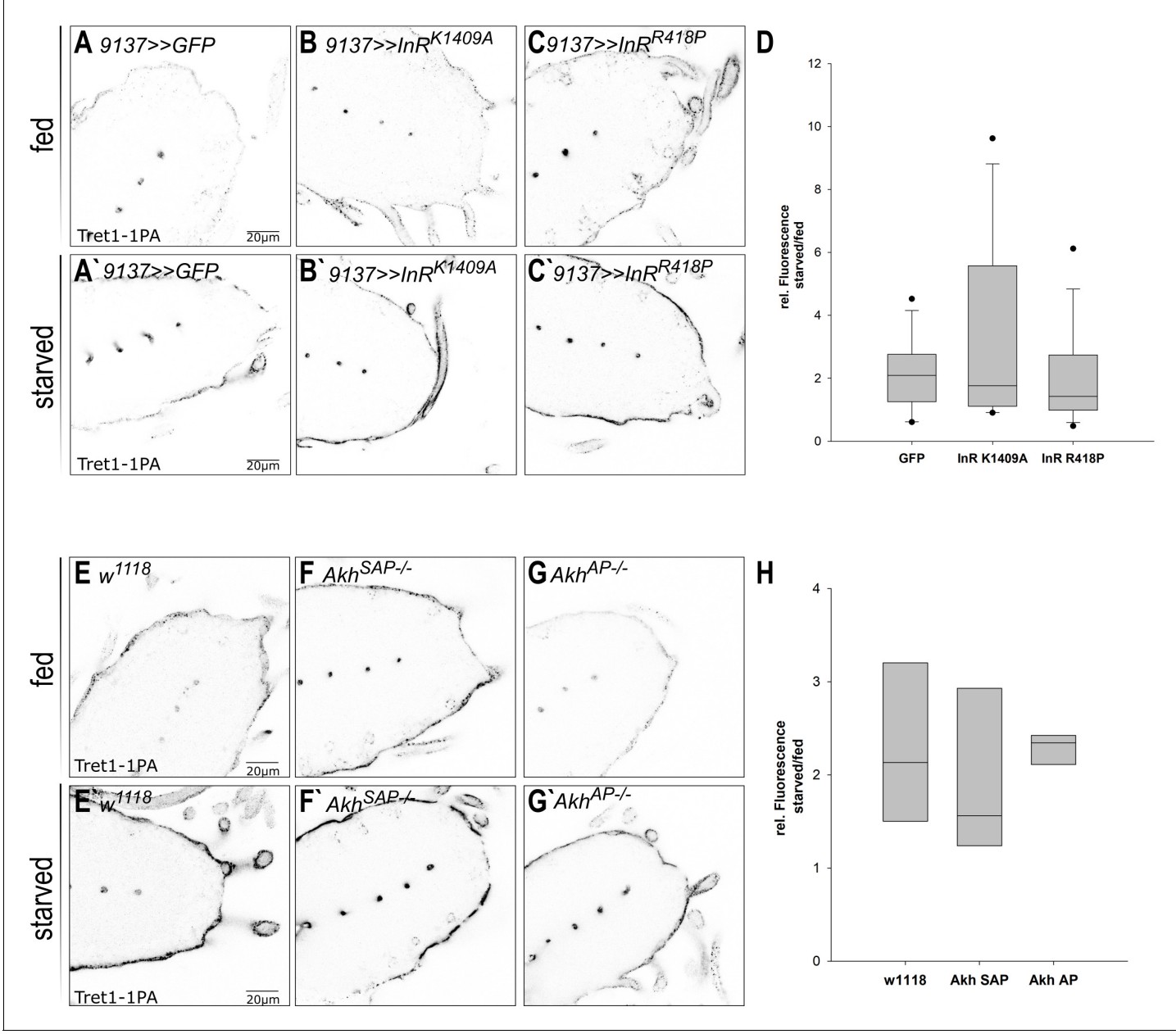

**Figure 6.** Tret1-1 regulation upon starvation is insulin and AKH independent. (A–C') Tret1-1 staining of the ventral nerve cord of starved and fed larvae expressing dominant-negative forms (InR$^{K1409A}$ or InR$^{R418P}$) of the insulin receptor (InR) in the perineurial and subperineurial glial cells. (B, B', C, C') Tret1-1 levels in fed and starved animals expressing InR dominant negative are indistinguishable from wild type. (B', C') Tret1-1 upregulation upon starvation is seen in all cases. (D) Quantification of Tret1-1 upregulation in animals expressing InR$^{K1409A}$ or InR$^{R418P}$. Shown is the ratio of relative Tret1-1 fluorescence intensity in the perineurial glial cells of starved versus fed animals. No significant differences between the genotypes are observed. N = 4, n = 12–16. (E–G') Tret1-1 staining of the ventral nerve cord of starved and fed wild-type and $Akh^{-/-}$ mutant animals ($Akh^{SAP}$ or $Akh^{AP}$). Tret1-1 levels in fed and starved mutant animals are indistinguishable from wild type. (F', G') Tret1-1 upregulation upon starvation can be seen in all mutants. (H) Quantification of Tret1-1 intensities of $Akh^{SAP}$ or $Akh^{AP}$. Shown is the ratio of relative Tret1-1 fluorescence intensity in the perineurial glial cells of starved versus fed animals. No significant differences are observed. N = 3, n = 5–8.

The online version of this article includes the following figure supplement(s) for figure 6:

**Figure supplement 1.** Tret1-1 regulation upon starvation is ALK-independent.

## Transforming growth factor β signaling regulates Tret1-1 expression

In *Drosophila*, both TGF-β/activin signaling and TGF-β/bone morphogenetic protein (BMP) signaling have been implicated in metabolic regulation (*Ballard et al., 2010*; *Ghosh and O'Connor, 2014*). The activin and BMP branches of TGF-β signaling share some components, like the type II receptors Punt (Put) and Wishful thinking (Wit) and the co-Smad Medea, while other components are specific to one or the other branch (reviewed in *Upadhyay et al., 2017*, *Figure 7K*).

Since Put has been implicated in regulating carbohydrate homeostasis, we asked if Put-dependent TGF-β signaling could also play a role in carbohydrate-dependent Tret1-1 regulation. Thus, we expressed dsRNA constructs against *put* in a glia-specific manner and analyzed Tret1-1 levels in the perineurial glial cells of fed and starved animals (*Figure 7B,B',C,C'*). Indeed, starvation-dependent upregulation of Tret1-1 was completely abolished upon *put* knockdown in glial cells using either *put$^{KK102676}$* or *put$^{GD2545}$*. Quantification shows no upregulation of Tret1-1 upon starvation in *put* knockdown animals (*Figure 7J*). In contrast, knockdown of *wit* using *wit$^{KK100911}$* did not affect Tret1-1 upregulation upon starvation (*Figure 7D,D'*). This data suggests that Put-dependent TGF-β signaling in glia is essential for starvation-induced upregulation of Tret1-1.

The activin-branch of TGF-β signaling has been shown to be important for sugar sensing and sugar metabolism in the adult fly as well as in larvae (*Chng et al., 2014*; *Ghosh and O'Connor, 2014*; *Mattila et al., 2015*). The type I receptor Baboon (Babo) is specific for the activin branch of TGF-β signaling (reviewed in *Upadhyay et al., 2017*; *Figure 7F,F'*). To test this, we silenced *babo* in glial cells using *babo$^{NIG8224R}$* that has been shown to efficiently abolish *babo* expression (*Hevia and de Celis, 2013*). Interestingly, in *babo* knockdown animals Tret1-1 expression is strongly upregulated upon starvation (*Figure 7J*), indicating that the Activin branch of TGF-β signaling is not implicated in Tret1-1 regulation.

This indicates that the BMP-branch of TGF-β signaling is implicated in *Tret1-1* regulation. To analyze its involvement, we knocked down the BMP-branch-specific type I receptors Thickveins (Tkv) and Saxophone (Sax) (reviewed in *Upadhyay et al., 2017*). Loss-of-function mutations in both *tkv* and *sax* are lethal, but Tkv overexpression can rescue *sax* loss-of-function; thus Tkv seems to be the primary type I receptor in the BMP-branch of TGF-β signaling (*Brummel et al., 1994*). Glia-specific knockdown of *sax* using *sax$^{GD50}$* or *sax$^{GD2546}$* did not show any differences in Tret1-1 regulation upon starvation compared to control knockdown animals (*Figure 7G,G',H,H'*). In contrast, knockdown of *tkv* using *tkv$^{KK102319}$* abolished Tret1-1 upregulation upon starvation, highlighting its importance for signaling (*Figure 7I,I'*).

## Glass-bottom boat-mediated TGF-β signaling induces Tret1-1 expression upon starvation

The BMP branch of TGF-β signaling can be activated by several ligands: glass-bottom boat (Gbb), decapentaplegic (Dpp), screw (Scw), and most likely Maverick (Mav) (reviewed in *Upadhyay et al., 2017*). Of those ligands, only Gbb has been implicated in regulating metabolic processes to date (reviewed in *Upadhyay et al., 2017*). *gbb$^{-/-}$* mutant animals show a phenotype that resembles the state of starvation, including reduced triacylglyceride storage and lower circulating carbohydrate levels (*Ballard et al., 2010*). It has previously been shown that overexpression of Gbb in the fat body leads to higher levels of circulating carbohydrates and thus the opposite of a starvation-like phenotype (*Hong et al., 2016*). Thus, to study their role in Tret1-1 regulation, we over-expressed Gbb or Dpp locally in the surface glial cells (9137-Gal4, perineurial and subperineurial glial cells) to avoid strong systemic impact that would counteract the effects of starvation (*Figure 8A–C*). In fed animals that express Gbb in the BBB cells, Tret1-1 expression is significantly upregulated in the perineurial glial cells (*Figure 8B,D*). This effect is specific to Gbb, since neither GFP-expressing control animals nor Dpp-expressing animals display this effect (*Figure 8A,B,D*). This shows that Gbb-dependent signaling does induce Tret1-1 upregulation. Furthermore, we analyzed the expression of Gbb, using antibodies (*Akiyama et al., 2012*). Gbb is found in the Tret1-1-expressing perineurial glial cells but seems to be also expressed in other glial cell types (most likely subperineurial glia and cortex glia) in the nervous system (*Figure 8G'*). Upon starvation, not only Tret1-1 but also Gbb expression is increased in the nervous system (*Figure 8H', I'*).

Taken together, the data reported here show that, upon starvation, increased levels of Gbb are found in the ventral nerve cord. Gbb activates the BMP-branch of TGF-β signaling in the perineurial

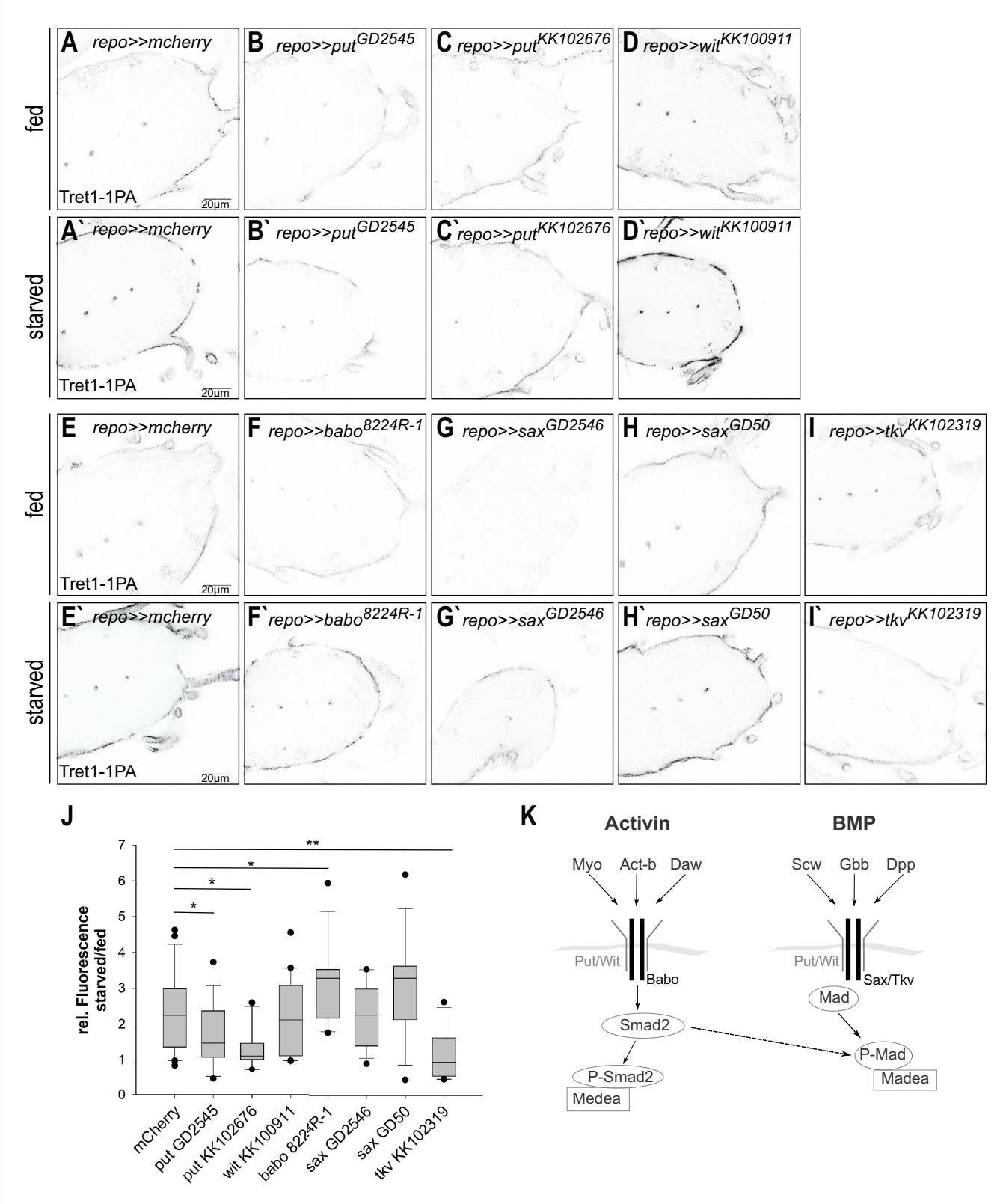

**Figure 7.** Tret1-1 upregulation upon starvation is BMP-mediated TGF-β signaling dependent. (A, A', I, I') Tret1-1 staining of the ventral nerve cord of starved and fed control (*repo>>mCherry-dsRNA*) animals and animals with a glial TGF-β knockdown. (B, B', C, C') Knockdown of the type I receptor Put in glial cells using two different dsRNA constructs (*put^GD2545* and *put^KK102676*) abolished Tret1-1 upregulation upon starvation. (D, D') Glia-specific knockdown of *wit* using *wit^KK100911* does not affect Tret1-1 upregulation upon starvation. (F, F') Glia-specific knockdown of the activin-branch-specific

*Figure 7 continued on next page*

*Figure 7 continued*

type II receptor Babo (using *babo^NIG8224R-1*) does not have any influence on Tret1-1 upregulation upon starvation (compare to control in E, E'). (G, G', H, H') The glia-specific knockdown of the BMP-branch-specific type II receptor Sax (using *sax^GD2546* and *sax^GD50*) does not influence Tret1-1 expression (compare to control in E, E'). (I, I') In contrast, glia-specific knockdown of the main BMP-branch-specific type II receptor Tkv (using *tkv^KK1023019*) abolishes Tret1-1 upregulation upon starvation (compare to control in E, E'). This indicates that signaling via the BMP branch of TGF-β signaling regulates Tret1-1 induction upon starvation. (J) Quantification of Tret1-1 upregulation upon starvation. Shown is the ratio of relative Tret1-1 fluorescence intensity in the perineurial glial cells of starved versus fed animals. N ≥ 4, n = 10–22. (K) Schematic representation of the two branches of the TGF-β signaling pathway.

glial cells, via the receptors Tkv (type I) and Put (type II), and induces Tret1-1 expression. Since it has been shown that mammalian GLUT1 is also upregulated upon hypoglycemia, it will be interesting to see whether TGF-β signaling is conserved as a pathway adapting carbohydrate transport to changes in nutrient availability.

## Discussion

The nervous system is separated from circulation by the BBB. This separation on the one hand protects the nervous system form circulation-derived harmful substances, but on the other hand necessitates efficient nutrient transport to ensure neuronal function. Since the nervous system mainly uses carbohydrates to meet its energetic demands, carbohydrates need to be taken up at a sufficient rate. We previously showed that the carbohydrate transporter Tret1-1 is specifically expressed in perineurial glial cells that surround the *Drosophila* brain and that glucose is taken up into the nervous system (*Volkenhoff et al., 2018*; *Volkenhoff et al., 2015*). Here, we investigated how Tret1-1-mediated carbohydrate uptake into the nervous system is adapted to the metabolic state of the animal to spare the nervous system from the effects of malnutrition. We show that Tret1-1 is a carbohydrate transporter that cannot only facilitate transport of trehalose as previously reported (*Kanamori et al., 2010*), but also of glucose (*Figure 5*). Upon chronic starvation, Tret1-1 protein levels are increased in the perineurial glial cells (*Figure 3*), boosting the glucose transport capacity in those cells (*Figure 5*). This effect reverts when Tret1-1 is knocked down (*Figure 5*), suggesting that Tret1-1 upregulation is crucial for adapting carbohydrate transport to adverse conditions. Lipoprotein particles were shown to be able to cross the BBB (*Brankatschk and Eaton, 2010*). An increase in lipid uptake and a partial switch to lipid usage for gaining energy might take place in addition to Tret1-1 upregulation and most likely upon longer phases of nutrient restrictions. Such metabolic changes would then most likely also have an effect on insulin signaling (*Brankatschk et al., 2014*).

Subcellular trafficking of Tret1-1, and its integration into the plasma membrane, is important for Tret1-1 homeostasis (*Figure 2*). Loss of Rab7 or Rab10 function has severe effects on Tret1-1 levels or localization (*Figure 2*). The intracellular accumulation of Tret1-1 induced by Rab10 silencing indicates that Tret1-1 cannot be properly delivered to the plasma membrane. Loss of Rab10 function in mammalian adipocytes induces perinuclear accumulation of GLUT4, suggesting regulatory parallels between Tret1-1 and GLUT4 (*Sano et al., 2007*). GLUT4 (*Slc2a4*) is weakly expressed in the mammalian BBB (*James et al., 1988*; *McCall et al., 1997*). Also, the two closest GLUT homologs of Tret1-1, GLUT6 and GLUT8, are regulated by subcellular trafficking from cytoplasmic storage vesicle to the plasma membrane (*Lisinski et al., 2001*). Both GLUT6 and GLUT8 are expressed in the mammalian brain, but their roles are unclear (*Doege et al., 2000a*; *Doege et al., 2000b*; *Ibberson et al., 2000*; *Reagan et al., 2002*).

We show that the *Tret1-1* promoter is induced upon starvation (*Figure 3*). This suggests that the *Tret1-1* locus harbors a starvation-responsive element. Tret1-1 levels are most likely regulated dependent on carbohydrate availability since animals feeding on sugar-only food do not show an upregulation of Tret1-1 (*Figure 4*). It has been reported that insulin-induced hypoglycemia leads to an upregulation of GLUT1 mRNA as well as protein in rat BBB-forming endothelial cells (*Kumagai et al., 1995*). In isolated rat brain microvessels, insulin-induced hypoglycemia also activates upregulation of GLUT1 protein levels and in addition an accumulation of GLUT1 at the luminal membrane (*Simpson et al., 1999*). In these rodent studies, GLUT1 upregulation was detected upon insulin injection that induces hypoglycemia. Under starvation conditions that lead to hypoglycemia in our experimental setup, however, insulin levels are strongly reduced. If under high-insulin conditions

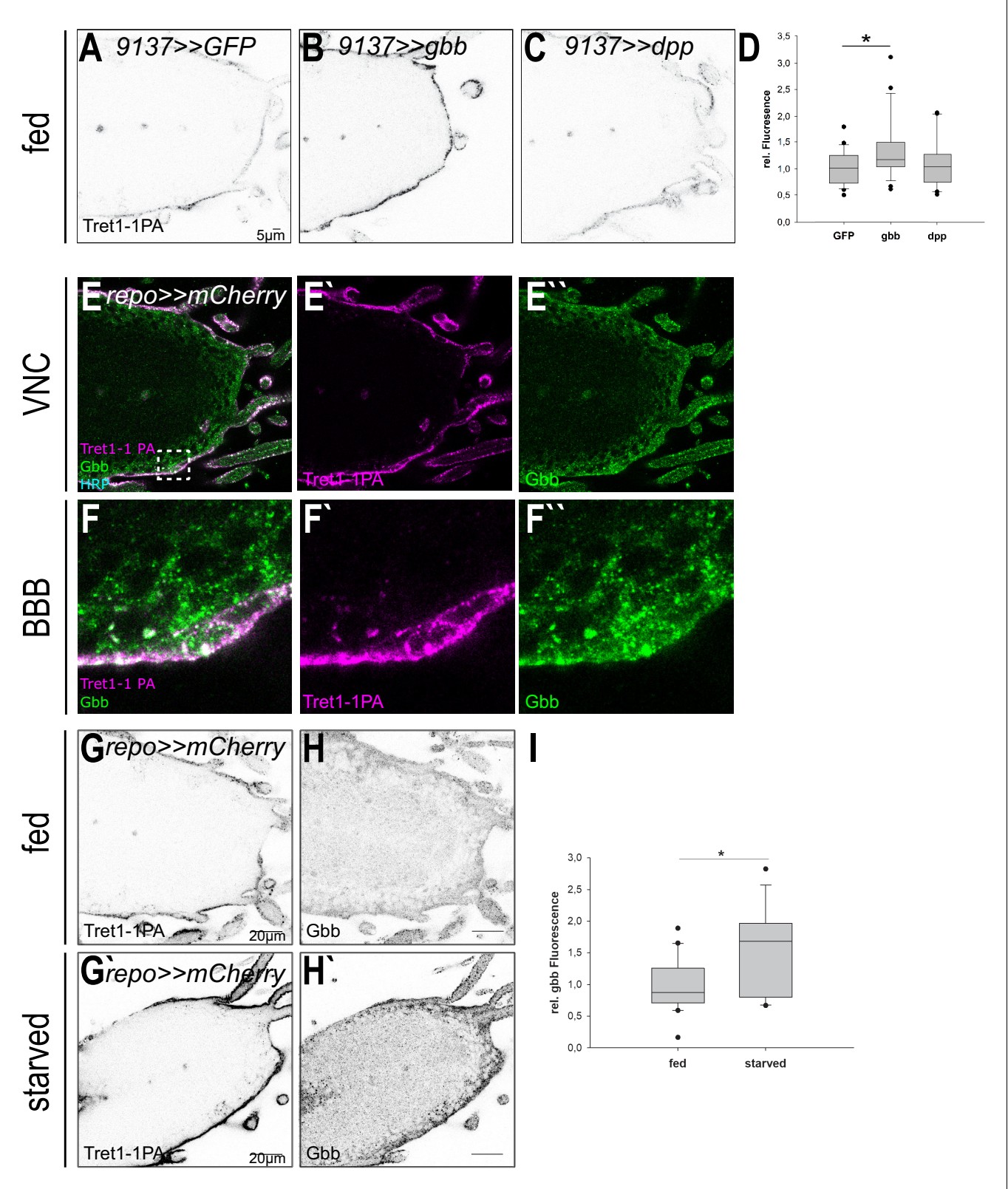

**Figure 8.** Tret1-1 upregulation depends on Gbb-mediated signaling. (**A–C**) Tret1-1 staining of the ventral nerve cord of fed control (9137>>GFP) animals or animals overexpressing either Gbb or Dpp in the perineurial and subperineurial glial cells (using 9137-Gal4). (**B**) Overexpression of Gbb induces increased Tret1-1 expression in fed animals. (**C**) Differently, overexpression of Dpp does not have any effect on the Tret1-1 expression. (**D**) Quantification of Tret1-1 upregulation upon overexpression of Gbb or Dpp in fed animals. The quantification shows Tret1-1 expression levels in the

*Figure 8 continued on next page*

*Figure 8 continued*

perineurial glial cells normalized to those in controls (9137>>GFP). N = 4; n = 20–25; (E–F'') Gbb is expressed in perineurial glial cells (coexpression with Tret1-1) and in other glial cell types, most likely subperineurial glial cells and cortex glial cells. (G–H') Upon starvation expression of Tret1-1 and Gbb is increased in the VNC. (I) Quantification of Gbb upregulation upon starvation in the brain. N = 5; n > 17.

GLUT1 levels are increased in mammals, this increase cannot be triggered by a loss of insulin. Along the same lines, the upregulation of Tret1-1 in perineurial glial cells we report here is independent of insulin signaling as well as AKH signaling (*Figure 6*). Thus, the regulatory mechanisms reported here may be conserved. This is especially interesting since aberrations in GLUT1 functionality or levels can cause severe diseases, such as GLUT1 deficiency syndrome or Alzheimer's (reviewed in *Koepsell, 2020*). Therefore, understanding the mechanisms that control the expression of carbohydrate transporters in the BBB-forming cells may be the basis for developing a treatment that allows to correct insufficient transporter expression in such diseases.

The induction of carbohydrate transport at the BBB upon hypoglycemia or starvation seems to be a mechanism that is required to spare the brain from the effects of malnutrition. It has previously been shown in mammals, as well as in flies, that the developing nervous system is protected from such effects to allow proper brain growth, while other organs undergo severe growth restriction. This process is called asymmetric intra-uterine growth restriction in humans or 'brain sparing' in model organisms (reviewed in *Lanet and Maurange, 2014*). In *Drosophila*, the mechanisms that underly the protection of the brain have been studied. Here, Jeb/ALK signaling in the neuroblast niche circumvents the need for insulin signaling to propagate growth (reviewed in *Lanet and Maurange, 2014*). Interestingly, Jeb/ALK signaling is not the basis for Tret1-1 upregulation in the perineurial glial cells, since glial ALK knockdown does not abolish Tret1-1 induction upon starvation (*Figure 6—figure supplement 1*).

TGF-β signaling has been shown to be involved in metabolic regulation in vertebrates and invertebrates (*Andersson et al., 2008*; *Bertolino et al., 2008*; *Ghosh and O'Connor, 2014*; *Zamani and Brown, 2011*). In *Drosophila*, the Activin-like ligand Dawdle (Daw) as well as the BMP ligand Glass-bottom boat (Gbb) have been implicated in metabolic regulation (reviewed in *Upadhyay et al., 2017*). Daw seems to be one of the primary players in the conserved ChREBP/MondoA-Mlx complex-dependent sugar-sensing pathway (*Mattila et al., 2015*). However, since the activin-like branch of TGF-β signaling does not play a role in Tret1-1 regulation, it does not seem to affect carbohydrate uptake into the nervous system. The BMP ligand Gbb, on the other hand, has been implicated in nutrient storage regulation. *gbb* mutants show expression defects of several starvation response genes (*Ballard et al., 2010*). Furthermore, the fat body of fed *gbb* mutants resembles that of starved wild-type animals by its nutrient storage and morphology (*Ballard et al., 2010*). Gbb seems to regulate nutrient storage in the fat body and control fat body morphology in a cell-autonomous manner. Additionally, since *gbb* mutants display increased nutrient uptake rates, *gbb* signaling also has systemic effects that are not yet completely understood (*Ballard et al., 2010*; *Hong et al., 2016*). We show here that upon starvation-elevated levels of Gbb signaling in the VNC induce an upregulation of Tret1-1 expression in perineurial glial cells (*Figure 8*). Gbb signals via Tkv and Put to regulate Tret1-1 expression upon starvation (*Figure 7*). Gbb was shown to act as proliferation factor in neuroblasts and also as a paracrine survival signal in perineurial glia (*Kanai et al., 2018*). However, we report here that Gbb is also expressed in other glial subtypes (*Figure 8*). Interestingly, it has been shown that BMP signaling induces transcriptional upregulation of GLUT1 in chondrocytes during murine skeletal development (*Lee et al., 2018*). Thus, TGF-β-dependent regulation of carbohydrate transport at the BBB may be based on the same mechanisms and consequently be evolutionarily conserved.

Interestingly, the transcription of the *Drosophila* sodium/solute cotransporter cupcake has also been shown to be upregulated upon starvation. Cupcake is expressed in some ellipsoid body neurons upon starvation and is essential for the ability of the animal to choose feeding on a nutritive sugar over feeding on a sweeter non-nutritive sugar after a period of nutrient deprivation (*Park et al., 2016*). Furthermore, several solute carrier family members have been shown to be regulated by carbohydrate availability in mouse cortical cell culture (*Ceder et al., 2020*). It will be very interesting to investigate whether such transcriptional upregulation is also mediated by TGF-β

signaling and whether TGF-β-mediated transcriptional regulation in the nervous system is a central mechanism that allows survival under nutrient shortage.

In summary, we report here a potentially conserved mechanism that protects the nervous system from effects of nutrient shortage by upregulation of carbohydrate transport at the BBB. This upregulation renders carbohydrate uptake more efficient and allows sufficient carbohydrate uptake even when circulating carbohydrate levels are low. In *Drosophila*, compensatory upregulation of Tret1-1 is regulated via Gbb and the BMP branch of TGF-β signaling. This mechanism is likely to be evolutionarily conserved, since mammalian GLUT1 has been shown to be regulated via BMP signaling in other tissues (*Lee et al., 2018*). These findings may serve as the basis of a future treatment against diseases caused by insufficient carbohydrate transport in the nervous system.

# Materials and methods

## Key resources table

| Reagent type (species) or resource | Designation | Source or reference | Identifiers | Additional information |
|---|---|---|---|---|
| Gene (*Drosophila melanogaster*) | Tret1-1 | | FBgn0050035 | |
| Genetic reagent (*D. melanogaster*) | jeb[KK111857] | Vienna *Drosophila* Resource Center | v103047; FBgn0086677; FBst0474909 | |
| Genetic reagent (*D. melanogaster*) | jeb[GD5472] | Vienna *Drosophila* Resource Center | v30800; FBgn0086677; FBst0458662 | |
| Genetic reagent (*D. melanogaster*) | Alk[GD42] | Vienna *Drosophila* Resource Center | v11446; FBgn0040505; FBst0450267 | |
| Genetic reagent (*D. melanogaster*) | put[KK102676] | Vienna *Drosophila* Resource Center | v107071; FBgn0003169; FBst0478894 | |
| Genetic reagent (*D. melanogaster*) | put[GD2545] | Vienna *Drosophila* Resource Center | v37279; FBgn0003169; FBst0461929 | |
| Genetic reagent (*D. melanogaster*) | wit[KK100911] | Vienna *Drosophila* Resource Center | v103808; FBgn0024179; FBst0475666 | |
| Genetic reagent (*D. melanogaster*) | sax[GD50] | Vienna *Drosophila* Resource Center | v42457; FBgn0003317; FBst0464598 | |
| Genetic reagent (*D. melanogaster*) | sax[GD2546] | Vienna *Drosophila* Resource Center | FBgn0003317 | |
| Genetic reagent (*D. melanogaster*) | tkv[KK102319] | Vienna *Drosophila* Resource Center | v105834; FBgn0003716; FBst0477660 | |
| Genetic reagent (*D. melanogaster*) | Rab10[GD13414] | Vienna *Drosophila* Resource Center | v28758; FBgn0015789; FBst0457628 | |
| Genetic reagent (*D. melanogaster*) | Rab10[GD16778] | Vienna *Drosophila* Resource Center | v46792; FBgn0015789; FBst0466897 | |
| genetic reagent (*D. melanogaster*) | Rab10[KK109210] | Vienna *Drosophila* Resource Center | v101454; FBgn0015789; FBst0473327 | |
| Genetic reagent (*D. melanogaster*) | Tret1-1[GD17787] | Vienna *Drosophila* Resource Center | v52360; FBgn0050035; FBst0469787 | |

*Continued on next page*

*Continued*

| Reagent type (species) or resource | Designation | Source or reference | Identifiers | Additional information |
|---|---|---|---|---|
| Genetic reagent (*D. melanogaster*) | Rab7[T22N] | Bloomington *Drosophila* Stock Center | 9778; FBgn0015795 | |
| Genetic reagent (*D. melanogaster*) | Rab10[T23N] | Bloomington *Drosophila* Stock Center | 9778; FBgn0015795; FBst0009778 | |
| Genetic reagent (*D. melanogaster*) | Rab7[EYFP] | Bloomington *Drosophila* Stock Center | 62545; FBgn0015795; FBst0062545 | |
| Genetic reagent (*D. melanogaster*) | Rab10[EYFP] | Bloomington *Drosophila* Stock Center | 62548; FBgn0015789; FBst0062548 | |
| Genetic reagent (*D. melanogaster*) | Rab19[EYFP] | Bloomington *Drosophila* Stock Center | 62552; FBgn0015793; FBst0062552 | |
| Genetic reagent (*D. melanogaster*) | Rab23[EYFP] | Bloomington *Drosophila* Stock Center | 62554; FBgn0037364; FBst0062554 | |
| Genetic reagent (*D. melanogaster*) | Rab7[TRIP.JF02377] | Bloomington *Drosophila* Stock Center | 27051; FBgn0015795; FBst0027051 | |
| Genetic reagent (*D. melanogaster*) | InR[K1409A] | Bloomington *Drosophila* Stock Center | FBgn0283499 | |
| Genetic reagent (*D. melanogaster*) | InR[R418P] | Bloomington *Drosophila* Stock Center | FBgn0283499 | |
| Genetic reagent (*D. melanogaster*) | UAS-dpp | Bloomington *Drosophila* Stock Center | 1486; FBgn0000490 | |
| Genetic reagent (*D. melanogaster*) | Cherry[dsRNA] | Bloomington *Drosophila* Stock Center | 35785; FBti0143385 | |
| Genetic reagent (*D. melanogaster*) | UAS-CD8-GFP | Bloomington *Drosophila* Stock Center | 30002 or 30003 | |
| Genetic reagent (*D. melanogaster*) | Akh[AP] | *Gáliková et al., 2015* doi: 10.1534/genetics. 115.178897 | FBal0319564 | |
| Genetic reagent (*D. melanogaster*) | Akh[SAP] | *Gáliková et al., 2015* doi: 10.1534/genetics. 115.178897 | FBal0319565 | |
| Genetic reagent (*D. melanogaster*) | babo[NIG8224R] | Japanese National Institute of Genetics | FBal0275907 | |
| Genetic reagent (*D. melanogaster*) | gliotactin-Gal4 | *Sepp et al., 2001* Doi: 10.1006/dbio. 2001.0411 | – | |
| Genetic reagent (*D. melanogaster*) | repo-Gal4 | *Sepp et al., 2001* Doi: 10.1006/dbio. 2001.0411 | – | |
| Genetic reagent (*D. melanogaster*) | 46 F-Gal4 | *Xie and Auld, 2011* Doi: 10.1242/dev.064816 | – | |
| Genetic reagent (*D. melanogaster*) | 9137-Gal4 | *DeSalvo et al., 2014* Doi: 10.3389/fnins. 2014.00346 | – | |
| Genetic reagent (*D. melanogaster*) | UAS-FLII[12]Pglu-700μδ6 | *Volkenhoff et al., 2018* Doi: 10.1016/j.jinsphys. 2017.07.010 | Maintained at S. Schirmeier lab | |
| Genetic reagent (*D. melanogaster*) | UAS-Gbb | P. Soba | – | |

*Continued on next page*

*Continued*

| Reagent type (species) or resource | Designation | Source or reference | Identifiers | Additional information |
|---|---|---|---|---|
| Genetic reagent (*D. melanogaster*) | UAS-RFP | S. Heuser | – | |
| Genetic reagent (*D. melanogaster*) | w[1118] | *Lindsley and Zimm, 1992* ISBN 9780124509900 | – | |
| Genetic reagent (*D. melanogaster*) | Tret1-1-stGFP | This paper | Maintained at S. Schirmeier | Tret1-1 promoter fusion to a nuclei-targeted GFP |
| Genetic reagent (*D. melanogaster*) | Tret1-1-Gal4 | This paper | Maintained at S. Schirmeier | Tret1-1 promoter induced Gal4 expression |
| Antibody | anti-Tret1-1 guinea pig polyclonal | *Volkenhoff et al., 2015* | Maintained at S. Schirmeier Lab | (1:50) |
| Antibody | anti-Laminin rabbit polyclonal | Abcam | ab11575 | (1:1000) |
| Antibody | anti-Repo mouse monoclonal | Developmental Studies Hybridoma Bank | 8D12 anti-Repo | (1:2) |
| Antibody | anti-GFP mouse monoclonal | Molecular Probes | A11120 | (1:1000) |
| Antibody | anti-GFP chicken polyclonal | Abcam | Ab92456 | (1:1000) |
| Antibody | anti-GFP JL-8 mouse monoclonal | Clontech | Cat. 632381 | (1:10000) WB |
| Antibody | anti-Tubulin mouse monoclonal | Developmental Studies Hybridoma Bank | 12G10 anti-alpha-tubulin | (1:80) WB |
| Antibody | anti-Apontic rabbit polyclonal | *Eulenberg and Schuh, 1997* | Gifted from Reinhard Schuh | (1:150) |
| Antibody | anti-Gbb mouse monoclonal | Developmental Studies Hybridoma Bank | GBB 3D6-24 | (1:20) |
| Recombinant DNA reagent | pBPGuw-stingerGFP (vector) | C. Klämbt | | |
| Recombinant DNA reagent | pBPGuwGal4 (vector) | Addgene | 17575 | |
| Recombinant DNA reagent | pGEM-He-Juel (vector) | S. Bröer | | |
| Sequence-based reagent | Forward primer_Tret1-1prom | This paper | PCR primers | CACCGGTCTCAAGCTCTCTTTTTTGCCTTACATATTTT |
| Sequence-based reagent | Reverse primer_Tret1-1prom | This paper | PCR primers | TGGGTAAGTTGGAGAGAGAG |
| Sequence-based reagent | Forward primer Tret1-1 PA | This paper | PCR primers | CGTCTAGAATGAGTGGACGCGAC |
| Peptide, recombinant protein | Reverse primer Tret1-1 PA | This paper | PCR primers | CGAAGCTTCTAGCTTACGTCACGT |
| Commercial assay or kit | pENTR/D- TOPO Cloning Kit | Thermo Fisher Scientific | K240020 | |
| Commercial assay or kit | mMESSAGE mMACHINE T7 Kit | Thermo Fisher Scientific | AM1344 | |
| Chemical compound, drug | $^{14}C_{12}$-trehalose | Hartmann Analytic, Braunschweig | #1249 | |

*Continued on next page*

*Continued*

| Reagent type (species) or resource | Designation | Source or reference | Identifiers | Additional information |
|---|---|---|---|---|
| Chemical compound, drug | $^{14}C_6$-glucose | Biotrend, Köln | #MC144-50 | |
| Chemical compound, drug | $^{14}C_6$-fructose | Biotrend, Köln | #MC1459-50 | |
| Chemical compound, drug | Rotiszint eco plus scintillation cocktail | Carl Roth | Art. No. 0016.3 | |
| Software, algorithm | SigmaPlot | Jadel | SPSS Inc | |
| Software, algorithm | Fiji | NIH | | |

## Fly stocks

Flies were kept at 25°C on a standard diet if not noted otherwise. The following fly stocks were used in this study: jeb[KK111857], jeb[GD5472], Alk[GD42], put[KK102676], put[GD2545], wit[KK100911], sax[GD50], sax[GD2546], tkv[KK102319], Rab10[GD13414], Rab10[GD16778], Rab10[KK109210], Tret1-1[GD17787](all fly stocks were obtained from VDRC Fly Center). Rab7[T22N], Rab10[T23N], Rab7[EYFP], Rab10[EYFP], Rab19[EYFP], Rab23[EYFP], Rab7[TRIP.JF02377], InR[K1409A], InR[R418P], UAS-dpp (BDSC 1486), Cherry[dsRNA] (BDSC 35785), UAS-CD8-GFP (BDSC 30002 or 30003) (all fly stocks were obtained from Bloomington *Drosophila* Stock Center). Akh[AP] and Akh[SAP] (*Gáliková et al., 2015*), babo[NIG8224R] (Japanese National Institute of Genetics), gliotactin-Gal4, repo-Gal4 (*Sepp et al., 2001*), 46 F-Gal4 (*Xie and Auld, 2011*), 9137-Gal4 (*DeSalvo et al., 2014*), UAS-FLII[12]Pglu-700μδ6 (*Volkenhoff et al., 2018*), UAS-Gbb (P. Soba), UAS-RFP (S. Heuser), w[1118] (*Lindsley and Zimm, 1992*).

## Creation of Tret1-1-Gal4 and Tret1-1-stinger-GFP flies

For creation of Tret1-1-Gal4 and Tret1-1-stinger-GFP flies, first the promotor region of *Tret1-1* was cloned from genomic DNA (forward primer_Tret1-1prom: CACCGGTCTCAAGCTCTCTTTTTTGCCTTACATATTTT, reverse primer_Tret1-1prom: TGGGTAAGTTGGAGAGAGAG) into the pENTR vector using the pENTR/D-TOPO Cloning Kit (Thermo Fisher Scientific). Via the gateway system, the promotor fragment was cloned either into the pBPGuwGal4 vector (Addgene #17575) or into pBPGuw-stingerGFP. Both clones were introduced into the 86Fb landing site via Φ integrase-mediated transgenesis (*Bischof et al., 2013*).

## Immunohistochemistry, SDS–PAGE, and western blotting

Third-instar larval brains or larval brains of animals that had been subjected to the larval starvation protocol were dissected and immunostained following standard protocols (*Volkenhoff et al., 2015*). Specimen were analyzed using the Zeiss 710 LSM or the Zeiss 880 LSM and the Airy Scan Module (Zeiss, Oberkochen, Germany). SDS–PAGE and western blotting were performed following published protocols (*Zobel et al., 2015*). Lysates were generated from 96 hr ± 3 hr old larval brains.

The following antibodies were used: guinea pig anti-Tret1-1 (1:50, *Volkenhoff et al., 2015*), rabbit anti-Laminin (1:1000, Abcam), mouse anti-Repo (1:2, Developmental Studies Hybridoma Bank), mouse anti-GFP (for immunohistochemistry: 1:1000, Molecular Probes; for western blotting: 1:10,000, Clontech), chicken anti-GFP (1:1000, Abcam), mouse anti-Tubulin (1:80, Developmental Studies Hybridoma Bank), rabbit anti-Apontic (1:150, *Eulenberg and Schuh, 1997*), mouse anti-Gbb (1:20, Developmental Studies Hybridoma Bank). As secondary conjugated antibodies, Alexa488- (1:1000), Alexa568- (1:1000), and Alexa647-coupled (1:500) antibodies were used (all from Thermo Fisher Scientific). For western blotting, goat anti-mouse HRP (Dianova, 1:7500) was used. HRP activity was detected using the ECL detection system kit (GE Healthcare) and the Amersham Imager 680 (GE Healthcare). Image analysis was performed using the Fiji plugin of ImageJ (1.52 p, java 1.8.0._172 64-bit, NIH, Bethesda, MD). N gives the number of independent experiments; n is the total number of animals analyzed.

## Larval starvation

Flies were kept overnight on standard food to stage the embryos. Fifty-six hours after larval hatching similar-sized larvae were collected, cleared from food, and transferred to different food conditions: standard food, water-soaked filter paper, or 10% sucrose in phosphate-buffered saline. They were kept for 40 hr on this condition before dissecting.

For fluorescent analysis, mean gray values of a region of interest (ROI) containing the entire tip of the ventral nerve cord were measured. The mean of values of seven single planes was taken of each brain and normalized to the size of the ROI. To obtain comparable values between experiments, the ratio of values received from starved animals to the mean of fed animals was calculated. Statistical analysis was performed using Sigma Plot software (Jadel). Differences were assessed by the Mann–Whitney rank sum test or t-test. p-values<0.05 were considered as significantly different.

## Measurement of glucose uptake

Larvae expressing $UAS\text{-}FLII^{12}Pglu\text{-}700\mu\delta6$ FRET glucose sensor under the control of $9137$-Gal4 and either $mCherry$-dsRNA or $Tret1\text{-}1$-dsRNA were kept on standard food or under starvation conditions following the larval starvation protocol. Larval brains were subsequently dissected in HL3 buffer (70 mM NaCl, 5 mM KCl, 20 mM $MgCl_2$, 10 mM $NaHCO_3$, 115 mM sucrose, 5 mM trehalose, 5 mM HEPES; pH 7.2; ca. 350 mOsm) and adhered to poly-D-lysine-coated coverslips. Coverslips were secured into a flow through chamber and mounted to the stage of a LSM880 confocal microscope (Zeiss, Oberkochen, Germany). The chamber was then connected to a mini-peristaltic pump (MPII, Harvard Apparatus) to allow buffer exchange.

Fluorescent images were acquired immediately after dissection using 20×/1.0 DIC M27 75 mm emersion objective (Zeiss, Oberkochen, Germany), with excitation 436/25 nm, beam splitter 455 nm, emission 480/40 nm (CFP channel); excitation 436/25 nm, beam splitter 455 nm, emission 535/30 nm (YFP channel). Each larval brain was imaged in a separate experiment (n = 10). After 2.5 min, HL3 buffer was exchanged for glucose buffer (HL3 supplemented with 10 mM glucose; pH 7.2) and replaced by HL3 again after a further 7.5 min.

For data analysis, a ROI containing the entire larval brain was selected and the mean gray value of all pixels minus background for each channel was calculated. Values were normalized to known minimum (HL3 buffer). Statistical and regression analysis of data obtained was performed using SigmaPlot software (Jandel). To determine glucose uptake rates, 10 time points 9 s after values rose above baseline levels were used to calculate the linear slope of each curve. Differences were assessed by the Mann–Whitney rank sum test (pairs). p-values<0.05 were considered as significantly different.

## *Xenopus* experiments

For isolation of oocytes, female *X. laevis* frogs (purchased from the Radboud University, Nijmegen, Netherlands) were anesthetized with 1 g/l of ethyl 3-aminobenzoate methanesulfonate and rendered hypothermic. Parts of ovarian lobules were surgically removed under sterile conditions. The procedure was approved by the Landesuntersuchungsamt Rheinland-Pfalz, Koblenz (23 177–07/A07-2-003 §6). Oocytes were singularized by collagenase treatment in $Ca^{2+}$-free oocyte saline (82.5 mM NaCl, 2.5 mM KCl, 1 mM $MgCl_2$, 1 mM $Na_2HPO_4$, 5 mM HEPES, pH 7.8, 2 mg/l gentamicin) at 28°C for 2 hr. The singularized oocytes were stored overnight at 18°C in $Ca^{2+}$-containing oocyte saline (82.5 mM NaCl, 2.5 mM KCl, 1 mM $CaCl_2$, 1 mM $MgCl_2$, 1 mM $Na_2HPO_4$, 5 mM HEPES, pH 7.8, 2 mg/l gentamicin). The procedure was described in detail previously (*Becker et al., 2014*).

For heterologous protein expression in *X. laevis* oocytes the *D. melanogaster* cDNA sequences of Tret1-1 isoform A was amplified via PCR from pUAST-Tret1-1-PA-3xHA plasmid (forward primer_-Tret1-1PA: CGTCTAGAATGAGTGGACGCGAC, reverse primer_Tret1-1PA: CGAAGCTTCTAGCTTACGTCACGT) and cloned into the pGEM-He-Juel vector using XbaI/ HindIII restriction sites. cRNA was produced by in vitro transcription using the mMESSAGE mMACHINE T7 Kit (Thermo Fisher Scientific). Oocytes of the stages V and VI were injected with 18 ng (for mass spectrometry) to 20 ng (for scintillation analysis) of cRNA, and measurements were carried out three to six days after cRNA injection.

To analyze the transport capacity by scintillation measurements, radioactive sugar substrates were generated using unlabeled sugar solutions of different concentrations in oocyte saline and adding $^{14}C$-labeled sugar at a concentration of 0.15 µCi/100 µl (for 0.3 mM–30 mM solutions) or 0.3

µCi/100 µl (for 100 mM and 300 mM solutions). $^{14}C_{12}$-trehalose was purchased from Hartmann Analytic, Braunschweig (#1249); $^{14}C_6$-glucose and $^{14}C_6$-fructose were purchased from Biotrend, Köln (#MC144-50 and 66 #MC1459-50). Six to eight oocytes were transferred into a test tube and washed with oocyte saline. Oocyte saline was removed completely, and 95 µl of the sugar substrate were added for 60 min. After incubation, cells were washed four times with 4 ml ice-cold oocyte saline. Single oocytes were transferred into Pico Prias scintillation vials (Perkin Elmer) and lysed in 200 µl 5% SDS, shaking at approximately 190 rpm for at least 30 min at 20°C–28°C. Three milliliters Rotiszint eco plus scintillation cocktail (Carl Roth) was added to each vial, and scintillation was measured using the Tri-Carb 2810TR scintillation counter (Perkin Elmer). Scintillation of 10 µl sugar substrate of each concentration with 200 µl 5% SDS and 3 ml Rotiszint eco plus scintillation cocktail served as a standard.

Substrate flux was calculated from the measured scintillation according to the respective standard measurements. For statistical analysis, the medium flux and standard error were calculated for oocytes expressing transport proteins and native oocytes and compared using a one-sided t-test or the Mann–Whitney rank test for analysis of non-uniformly distributed samples. Determination of the net-flux was performed by subtracting the medium flux of native oocytes from one test series from each measurement of the same test series and calculating the medium flux and standard error.

## Acknowledgements

We are grateful to M Brankatschk for fly stocks. We thank Astrid Fleige for help with cloning and western blots. We are grateful to C Klämbt for discussions and critical reading of the manuscript. The work was supported by grants of the DFG to SS (SFB1009, SCHI 1380/2–1).

## Additional information

### Funding

| Funder | Grant reference number | Author |
|---|---|---|
| Deutsche Forschungsgemeinschaft | SFB1009 | Stefanie Schirmeier |
| Deutsche Forschungsgemeinschaft | SCHI 1380/2-1 | Stefanie Schirmeier |

The funders had no role in study design, data collection and interpretation, or the decision to submit the work for publication.

### Author contributions

Helen Hertenstein, Conceptualization, Data curation, Formal analysis, Investigation, Visualization, Writing - original draft, Writing - review and editing; Ellen McMullen, Formal analysis, Investigation, Visualization, Methodology; Astrid Weiler, Formal analysis, Investigation, Visualization; Anne Volkenhoff, Formal analysis, Investigation, Methodology; Holger M Becker, Conceptualization, Formal analysis, Supervision, Investigation, Methodology; Stefanie Schirmeier, Conceptualization, Resources, Data curation, Formal analysis, Supervision, Funding acquisition, Investigation, Visualization, Writing - original draft, Project administration, Writing - review and editing

### Author ORCIDs

Holger M Becker (iD) http://orcid.org/0000-0002-2700-6117
Stefanie Schirmeier (iD) https://orcid.org/0000-0001-8431-9593

### Decision letter and Author response

Decision letter https://doi.org/10.7554/eLife.62503.sa1
Author response https://doi.org/10.7554/eLife.62503.sa2

## Additional files

### Supplementary files
• Transparent reporting form

### Data availability
The full imaging raw data is available at https://dx.doi.org/10.17879/37089751811.

The following dataset was generated:

| Author(s) | Year | Dataset title | Dataset URL | Database and Identifier |
|---|---|---|---|---|
| Hertenstein H, McMullen E, Weiler A, Volkenhoff A, Becker HM, Schirmeier S | 2021 | Starvation-induced regulation of carbohydrate transport at the blood-brain barrier is TGF-$\beta$-signaling dependent | https://miami.uni-muenster.de/Record/7024836e-8e03-4e62-8202-b66b42a6dda0 | Miami, 10.17879/37089751811 |

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
