## [Decision Letter]

**Acceptance summary:**

The brain is protected from nutrient deprivation. Here, the authors investigate the possible mechanism by which this protection might occur. They make a case for starvation resulting in the release of the transforming growth factor pathway ligand. This activates an arm of the pathway in the perineurial glia, which acts as a blood-brain barrier in flies. This in turn results in the transcriptional upregulation of a carbohydrate transporter. This transporter is capable of transporting glucose and that its capacity to do so enhances upon starvation. They postulate that this is the mechanism by which the brain might be protected from nutrient deprivation.

**Decision letter after peer review:**

Thank you for submitting your article "Starvation-induced regulation of carbohydrate transport at the blood-brain barrier is TGF-β-signaling dependent" for consideration by *eLife*. Your article has been reviewed by 2 peer reviewers, and the evaluation has been overseen by K VijayRaghavan as the Senior Editor and Reviewing Editor. The reviewers have opted to remain anonymous.

The reviewers have discussed the reviews with one another and the Reviewing Editor has drafted this decision to help you prepare a revised submission.

This paper addresses the interesting question of how the brain reacts differently to nutrient deprivation as compared to the periphery, variously referred to as brain sparing or the self brain hypothesis by some researchers. Using *Drosophila* larvae, the paper shows that the gene Tret1-1 is transcriptionally upregulated in perineurial BBB cells upon sugar starvation, the gene product is localized in vesicles that is used to transport it to the membrane in a Rab dependent manner. The gene product can transport glucose in *Xenopus* oocyte system and upregulation of the gene through starvation results in increased transport of glucose in the brain, as shown by in vivo glucose sensor. In the second phase of the paper, they show that this starvation induced upregulation of Tret1-1 is not dependent on insulin, akh, or Alk signaling, but rather by a specific branch of TGFb signaling, namely the one driven by the ligand Gbb.

The strength of the paper is the admirable usage of different techniques to get at a specific issue at a deep resolution, from molecular genetics to biochemistry to neuronal imaging. A potential disadvantage is how rigorous one can or should go in any one of these areas.

Both the reviewers outline some points meant not as strict requirements, but rather for consideration for going a bit deeper. As you can see, most of their points overlap. It will be valuable if the manuscript is revised taking their views into consideration/

*Reviewer 1:*

1. If you downregulate Tret1-1, do you eliminate glucose uptake increase upon starvation? What´s the phenotype in terms of being more or less resistant to starvation?

2. They test insulin, akh and Alk signaling and show these are not involved. I´m just curious why they have not tested/shown TOR (target of rapamycin). I realize there are many signaling pathways to test, but this was such an obvious candidate in terms of nutrient signaling, especially based on studies on yeast where TOR is involved in putting different types of nutrient transporters to the cell surface.

3. What is known about Gbb expression? The manuscript is unclear on this. They overexpress Gbb in BBB (both peri and subperineurial glia), and they say that "moderate levels of Gbb are produced by an unknown source probably locally in the subperineurial glial cells" (page 13, lines 388-389). Is Gbb expressed in subperineurial glia? Is it upregulated upon starvation?

4. It would be good if they could also show, as an internal control, that some aspect of the above signaling events in the periphery is different than in the brain, e.g. some other sugar transporter or signaling components being downregulated in peripheral tissue upon starvation.

*Reviewer 2:*

The central case of this story is that the increase in Tret-1 expression is what protects the brain from conditions of starvation. In the manuscript, the evidence to suggest this is indirect. I expected that the authors would have shown the following:

1. In the absence of Tret-1, there is a lack of (or an attenuation in) the protection of the brain from the starved condition. How they assay the protection of the brain could be via size of the brain, reactivation of NBs, or any other means they thought robust and sensitive.

2. In the same condition – absence of Tret-1 and in conditions of starvation – that there is an attenuation of a lack of Glucose transport as measured by the glucose sensor in the blood brain barrier.

The authors seem to suggest that in the starved condition not only does Tret-1 transcription increase, but it also gets re-localized from intracellular vesicles to the plasma membrane. The subcellular localization is interesting because I presume for Tret-1 to act as a transporter of glucose it must localize to the membrane. However, this point is not addressed in the rest of the story (I am not suggesting this as a necessary experiment), and the altered localization was not obvious to me. Could the authors quantify the localisation changes in some manner? And maybe discuss how they think starvation might result in the relocalisation? Or is it possible that membrane localization is present in both conditions, but the increased levels of Tret-1 in the starved condition just shows membrane localization more clearly?

Was there a reason why the authors did not do a quantitative PCR to determine transcriptional changes in tret-1 in response to starvation? Instead, they use a reporter-based assay for this. They do show a Western in figure 3, but I am concerned that the level of tubulin are very different in the fed and starved conditions. Particularly because in 3B they use a ratiometric quantification to show enhanced levels of the protein, the ratiometric quantification will show an exaggerated difference in the starved condition if the tubulin levels are low.

---

## [Author Response]

Reviewer 1:1. If you downregulate Tret1-1, do you eliminate glucose uptake increase upon starvation?

To answer this question, we expressed Tret1-1dsRNA in glial cells and measured glucose uptake into the brains of those animals under standard conditions and upon starvation. Tret1-1 knockdown abolishes the increase of glucose uptake upon starvation. We added this data to the manuscript (Figure 5 and corresponding text). The glucose uptake under fed conditions increases slightly upon Tret1-1 knockdown. This effect is most likely caused by overcompensation via other sugar transporters that are expressed in the BBB (McMullen et al., 2021). It seems however that such compensation cannot rescue glucose uptake in starved animals. This increase in glucose uptake, thus, is specifically caused by an increased expression of Tret1-1.

What´s the phenotype in terms of being more or less resistant to starvation?

We are sorry for being unclear about this. Animals which are completely starved after an age of 70 hours (25 °C) do not grow anymore but nevertheless survive and give rise to adults. Therefore, we start starving larvae approx. 14 hours before this developmental timepoint, since this period is critical for development. Wild type animals can survive starvation for 40 h at this stage (see also new scheme in Figure 1), although they do not grow during this period and just start growing again if refed. If the starvation is prolonged, the animals die. Animals that are starvation susceptible, like jeb knockdown larvae, do not survive this starvation paradigm, a phenotype that we termed “starvation susceptibility”. We explained this better in the text now.

2. They test insulin, akh and Alk signaling and show these are not involved. I´m just curious why they have not tested/shown TOR (target of rapamycin). I realize there are many signaling pathways to test, but this was such an obvious candidate in terms of nutrient signaling, especially based on studies on yeast where TOR is involved in putting different types of nutrient transporters to the cell surface.

Thank you for this suggestion. In the current study we aimed to analyze how Tret1-1 upregulation upon starvation is regulated, since we found that this occurs via TGF-β signaling, we did not test all other pathways that could potentially be involved any more.

Signaling via TOR is indeed an obvious candidate. Thus, to answer the question we knocked down TOR using two different dsRNA constructs. We did not see any impairment of Tret1-1 upregulation in such animals, indicating that TOR is not involved in Tret1-1 regulation. See Author response image 1. Since the result was negative, we did not include the data in the manuscript.

**Author response image 1. sa2fig1:** Tret1-1 upregulation upon starvation is independent of TOR signaling. Quantified is the ratio of Tret1-1 fluorescence intensities in starved larval VNCs vs. VNCs of fed animals. Knockdown of TOR expressing using either TOR^33951^ or TOR^35578^ in glial cells shows a comparable upregulation of Tret1-1 to the control. N=5 n >9.

3. What is known about Gbb expression? The manuscript is unclear on this. They overexpress Gbb in BBB (both peri and subperineurial glia), and they say that "moderate levels of Gbb are produced by an unknown source probably locally in the subperineurial glial cells" (page 13, lines 388-389). Is Gbb expressed in subperineurial glia? Is it upregulated upon starvation?

There is an antibody against Gbb available that was previously used to stain Gbb in NMJs (Akiyama et al., 2012; James et al., 2014). We used this antibody now to stain brains for Gbb. The staining is weak, but specific. Gbb seems to be expressed in all cell types of the nervous system. Upon starvation Gbb is upregulated in the brain. We included this data into the manuscript (Figure 8 and corresponding text).

Further attempts to identify the cell type that is crucial for Tret1-1 induction upon starvation failed unfortunately. We overexpressed Gbb in different glial cell types (also just SPGs or just PGs), but unfortunately these experiments led to (semi-)lethality of the animals. This phenotype is most likely caused by developmental problems caused by the overexpression of Gbb in the brain or other tissues. Gbb is important during development. The used Gal4 driver lines also show expression in other cell types outside the central nervous system and also early in development. This additional expression most likely causes the (semi)-lethality and thus does not allow analyses of the effect of glial subtype specific overexpression of Gbb.

Next, we attempted to identify the cell type from which Gbb was secreted using Gbb knockdown (HMS01243) in either perineurial or subperineurial glial cells. However, knockdown of Gbb in these cell types did not change Tret1-1 upregulation upon starvation. This most likely means that the signal that induces Tret1-1 upregulation does not originate from a single cell type, which would go along with the expression and starvation-induced induction of Gbb in several glial cell types. Alternatively, different cell types could compensate for the loss of Gbb signal from one cell type. We were unfortunately not able to test a panglial knockdown of Gbb due to restricted access to the lab. Since these results unfortunately still do not answer the question where the signal comes from, we did not include them in the manuscript.

4. It would be good if they could also show, as an internal control, that some aspect of the above signaling events in the periphery is different than in the brain, e.g. some other sugar transporter or signaling components being downregulated in peripheral tissue upon starvation.

Attempts to study Tret1-1 expression in muscle upon starvation did not give coherent results. Due to the very thick tissue of the muscles and the extremely big cells, the fluorescence intensities of Tret1-1 vary a lot within a Z-stack. Analyzing and comparing the intensity of Tret1-1 staining in between animals and in between animals with different genetic backgrounds was therefore not possible. Further attempts to study Tret1-1 regulation in peripheral tissues were impossible due to the restricted access to the lab, that forced us to prioritize certain experiments over others. In *Drosophila*, hardly anything is known about the expression of other sugar transporters in the periphery. Thus, analyzing the expression patterns of other putative sugar transporters and their regulation in the periphery upon starvation would be a big endeavor and was unfortunately not possible within the timeframe of this revision.

Reviewer 2:The central case of this story is that the increase in Tret-1 expression is what protects the brain from conditions of starvation. In the manuscript, the evidence to suggest this is indirect. I expected that the authors would have shown the following:1. In the absence of Tret-1, there is a lack of (or an attenuation in) the protection of the brain from the starved condition. How they assay the protection of the brain could be via size of the brain, reactivation of NBs, or any other means they thought robust and sensitive.

Thank you for this suggestion. We tried to measure the size of the brains of starved wild type and Tret1-1- knockdown animals. We did this using comparable confocal slices of each brain and measuring the area of the brain (2D) in those slices. When doing this the difference between control brains and Tret1-1 knockdown brains is not significant, even though there is a tendency towards being smaller in Tret1-1 knockdown brains. To analyze brain differences in brain volume (3D), which most likely will show a bigger difference, we would need many additional scans of starved brains for both genotypes. Due to restricted access to the lab, we were unfortunately not able to conduct those additional experiments. These pandemic-induced restrictions unfortunately also prevented us from analyzing NB reactivation.

2. In the same condition – absence of Tret-1 and in conditions of starvation – that there is an attenuation of a lack of Glucose transport as measured by the glucose sensor in the blood brain barrier.

To answer this question, that was also raised by reviewer I, we expressed Tret1-1dsRNA in glial cells and measured glucose uptake into the brains of those animals under standard conditions and upon starvation. Tret1-1 knockdown abolishes the increase of glucose uptake upon starvation. We added this data to the manuscript (Figure 5 and corresponding text). The glucose uptake under fed conditions increases slightly upon Tret1-1 knockdown. This effect is most likely caused by overcompensation via other sugar transporters that are expressed in the BBB (McMullen et al., 2021). It seems however that such compensation cannot rescue glucose uptake in starved animals. This increase in glucose uptake, thus, is specifically caused by an increased expression of Tret1-1.

The authors seem to suggest that in the starved condition not only does Tret-1 transcription increase, but it also gets re-localized from intracellular vesicles to the plasma membrane. The subcellular localization is interesting because I presume for Tret-1 to act as a transporter of glucose it must localize to the membrane. However, this point is not addressed in the rest of the story (I am not suggesting this as a necessary experiment), and the altered localization was not obvious to me. Could the authors quantify the localisation changes in some manner? And maybe discuss how they think starvation might result in the relocalisation? Or is it possible that membrane localization is present in both conditions, but the increased levels of Tret-1 in the starved condition just shows membrane localization more clearly?

Thank you for this remark. Several attempts (TIRF microscopy, expansion microscopy) were performed to analyze a change in Tret1-1 localization upon starvation in more detail. The small size of perineurial glia make this analysis very difficult. Mammalian GLUT4 is stored in storage vesicles in the cytoplasm of adipose cells. Upon increased insulin levels these vesicles are recruited towards the membrane and GLUT4 is mostly localized at the membrane (Klip et al., 2019). Tret1-1 upregulation upon starvation is independent of insulin signaling, however. The general increase of Tret1-1 expression upon starvation might indirectly increase the amount of Tret1-1 at the plasma membrane independently of a relocation. Thus, we cannot exclude that under starvation conditions the same portion of Tret1-1 protein localizes to the plasma membrane as under normal fed conditions. Tret1-1 colocalization with Rab GTPases, however, might suggest a similar translocation as seen for mGLUT4. We rephrased this part in the manuscript and discussion to account for the hypothetical character of the translocation idea.

Was there a reason why the authors did not do a quantitative PCR to determine transcriptional changes in tret-1 in response to starvation? Instead, they use a reporter-based assay for this. They do show a Western in figure 3, but I am concerned that the level of tubulin are very different in the fed and starved conditions. Particularly because in 3B they use a ratiometric quantification to show enhanced levels of the protein, the ratiometric quantification will show an exaggerated difference in the starved condition if the tubulin levels are low.

Quantitative PCR is a very delicate technique to analyze transcriptional changes. Tret1-1 is specifically expressed in perineurial glia that represents a very small number of cells in the brain. Therefore, it is not trivial to detect Tret1-1 mRNA quantitatively in the mRNA of brain samples. Since this approach would likely have required a lot of optimization, we decided to use another approach. We agree, however, that our western blots might not be optimal. Thus, we now added another analysis to verify the results of the western blots. We quantified stgGFP (in Tret1-1>stgGFP animals) in individual nuclei and normalized it to DAPI, since it seems very unlikely that the DNA content of a cell would change upon starvation. This technique is still ratiometric, but both independent approaches show that Tret1-1 is upregulated on a transcriptional level. Thus, we are now very confident that our observation is true. We included this data in Figure 3 and the corresponding text.